# Identifying modules of cooperating cancer drivers

Michael I Klein[1,2] iD, Vincent L Cannataro[3,4], Jeffrey P Townsend[1,4,5], Scott Newman[2], David F Stern[5,6,*,†] iD & Hongyu Zhao[1,4,5,**,†] iD

## Abstract

Identifying cooperating modules of driver alterations can provide insights into cancer etiology and advance the development of effective personalized treatments. We present Cancer Rule Set Optimization (CRSO) for inferring the combinations of alterations that cooperate to drive tumor formation in individual patients. Application to 19 TCGA cancer types revealed a mean of 11 core driver combinations per cancer, comprising 2–6 alterations per combination and accounting for a mean of 70% of samples per cancer type. CRSO is distinct from methods based on statistical co-occurrence, which we demonstrate is a suboptimal criterion for investigating driver cooperation. CRSO identified well-studied driver combinations that were not detected by other approaches and nominated novel combinations that correlate with clinical outcomes in multiple cancer types. Novel synergies were identified in *NRAS*-mutant melanomas that may be therapeutically relevant. Core driver combinations involving *NFE2L2* mutations were identified in four cancer types, supporting the therapeutic potential of NRF2 pathway inhibition. CRSO is available at https://github.com/mikekleinsgit/CRSO/.

**Keywords** cancer etiology; driver-gene combinations; multi-gene biomarkers; patient stratification; precision oncology
**Subject Categories** Cancer; Computational Biology; Molecular Biology of Disease
**Mol Syst Biol. (2021) 17: e9810**

## Introduction

Carcinogenesis usually involves the deregulation of multiple genetic pathways. Most recent estimates are that this deregulation requires 2–8 "hits" to enable a precursor cell to become neoplastic (Vogelstein *et al*, 2013; Tomasetti *et al*, 2015; Anandakrishnan *et al*, 2019). Identifying a small number of driver mutations among an excess of passenger mutations is challenging. Nevertheless, many methods have been developed. Generally, these methods identify genes mutated significantly more than the background mutation rate would predict and their output is a list of significant genes or regions found across a whole cohort. Well-known examples include MutSigCV (Lawrence *et al*, 2013) and dNdScv (Martincorena *et al*, 2017) for identification of significantly mutated genes (SMGs), and GISTIC2 (Mermel *et al*, 2011) for identification of significant somatic copy number variations (SCNVs).

Although useful in identifying cancer genes, these statistical enrichment methods do not attempt to identify co-occurring mutations. For example, activating mutations in *KRAS/NRAS* are frequently accompanied by loss of function of *CDKN2A/B* in melanoma, lung, pancreatic, and other cancers. A major reason for co-occurrence is that loss of the G1/S checkpoint (*CDKN2A/B*) is necessary to avoid oncogene-induced senescence caused by *KRAS* oncogenic signaling (Aguirre *et al*, 2003; Courtois-Cox *et al*, 2008; Schuster *et al*, 2014). Thus, mutations in the G1/S checkpoint pathway and in the RAS pathway often cooperate to produce a pro-growth phenotype. Many methods have been developed that identify mutually exclusive candidate driver genes (Miller *et al*, 2011; Leiserson *et al*, 2013; Leiserson *et al*, 2015; Wu *et al*, 2015; Hou *et al*, 2016; Bokhari & Arodz, 2017; Gao *et al*, 2017), but there are comparatively few that identify functionally relevant modules of co-occurring gene alterations in individual patients. This extra layer of biological information and its possible relevance to therapy is not captured by any of the above methods, and it would be beneficial to develop new approaches that identify groups of cooperating mutations.

From a biological perspective, co-occurrence of driver alterations clearly does occur (Ulz *et al*, 2016; Wang *et al*, 2017; VanderLaan *et al*, 2017; Kim *et al*, 2019), and appears to be a requirement for most carcinogenesis events, as evidenced by the insufficiency of *BRAF* and *RAS* hotspot mutations to transform benign colon polyps and nevi into invasive carcinoma (Pollock *et al*, 2003; Juárez *et al*, 2017). However, statistical approaches to identifying co-occurrence have given mixed results. For example, Canisius *et al* (2016) concluded that, after accounting for patient-specific mutation

1 Program in Computational Biology and Bioinformatics, Yale University, New Haven, CT, USA
2 Bioinformatics R&D, Sema4, Stamford, CT, USA
3 Department of Biology, Emmanuel College, Boston, MA, USA
4 Department of Biostatistics, Yale School of Public Health, New Haven, CT, USA
5 Yale Cancer Center, Yale University, New Haven, CT, USA
6 Department of Pathology, Yale School of Medicine, New Haven, CT, USA
  *Corresponding author. Tel: +1 203 785 4832; E-mail: df.stern@yale.edu
  **Corresponding author. Tel: +1 203 785 3613; E-mail: hongyu.zhao@yale.edu
  †These authors contributed equally to this work

frequencies, there is no evidence of statistically significant co-occurrence of somatic mutations in cancer. Conversely, Mina *et al* (2017) did identify pairwise oncogenic synergies in multiple cancer types based on a model of sequential evolution using their SELECT algorithm. Using mathematical modeling of cancer evolution, Mina *et al* argue that statistically significant co-occurrence emerges only by conditioning on the sequence of alterations, consistent with the lack of traditional co-occurrence demonstrated by Canisius *et al* (2016). Zhang *et al* (2014) found evidence of statistically significant co-occurrence at the level of the pathway but did not investigate at the gene level. Thus, current approaches can identify pairwise combinations of driver mutations in a subset of individuals. However, methods to detect combinations of three or more drivers and methods that attempt to identify driver combinations within every sample in a cohort (or at least within a large proportion of samples) are needed.

Here, we describe Cancer Rule Set Optimization (CRSO), a method to identify modules of cooperating alterations that are essential and collectively sufficient to drive cancer in individual patients. CRSO is developed as part of a theoretical framework that assumes the existence of specific combinations of two or more alterations called *rules* that cooperatively drive cancer if found in the same host cell. Rules are assumed to be minimally sufficient, meaning that exclusion of any of the alterations renders the remaining collection of alterations insufficient to drive cancer. CRSO seeks to find a collection of rules called a *rule set* that represents all of the unique minimal combinations that can explain all of the given tumors in the study population, i.e., every sample is required to harbor all of the alterations in at least one of the rules. CRSO is robust to heterogeneous mutational rates within the cohort as it uses alteration-specific passenger probabilities that reflect how likely specific observations would have occurred by chance. The output of CRSO can provide biological insights to cancer causation, as well as infer the likely driver combinations in individual patients. We show that CRSO can identify known and novel combinations of driver alterations in 19 tissue types from The Cancer Genome Atlas (TCGA) (Cancer Genome Atlas Network, 2013; Tomczak *et al*, 2015) and that some of these combinations correlate with clinical outcomes.

## Results

### CRSO overview

CRSO finds combinations of genomic alterations (referred to as *events*) that are predicted to cooperate to drive cancer in individual patients. The inputs into CRSO are two event-by-sample matrices: **D** and **P**. **D** is a binary alteration matrix, such that $D_{ij} = 1$ if event $i$ occurs in sample $j$, and 0 otherwise. **P** is a continuous valued penalty matrix, where $P_{ij}$ is the negative log of the probability of event $i$ occurring in sample $j$ by chance, i.e., as a passenger event. Three primary event types were considered: coding mutations identified as candidate drivers by dNdScv (Martincorena *et al*, 2017), as well as copy number amplifications and deletions identified as candidate drivers by GISTIC2 (Mermel *et al*, 2011). The version of CRSO presented here was restricted to SCNVs and coding mutations because candidate drivers for these alteration classes can be readily generated by statistical enrichment methods. However,

CRSO models can include additional alteration types, such as gene fusions and aneuploidies.

In order to calculate passenger probabilities, mutations and copy number variations were first represented as a categorical event-by-sample matrix, **M**. $M_{ij}$ can take one of several values called *observation types*, that depend on the type of event $i$. Mutation events take values in the set $\{Z, HS, L, S, I\}$, corresponding to wild type, hotspot mutation, loss mutation, splice site mutation, or in-frame indel (Materials and Methods). Amplification events take values in $\{Z, WA, SA\}$, corresponding to wild type, weak amplification, and strong amplification, respectively. Similarly, deletion events take values in $\{Z, WD, SD\}$, corresponding to wild type, weak deletion (hemizygous), and strong deletion (homozygous), respectively. Two additional event types, referred to as hybrid mutDels and mutAmps, were defined to represent genes that were identified by both dNdScv and GISTIC2 in the same tumor type (Materials and Methods). Figure 1A shows an example of the input matrices using a miniature dataset that was extracted from TCGA melanoma (SKCM) data for the purposes of illustration.

The CRSO model defines a *rule* as a collection of two or more events that drive tumors harboring all of the events in the collection. Rules are defined to be minimally sufficient, meaning that any strict subset of events within the rule are insufficient to cause cancer. A *rule set* is defined as a collection of rules that account for all minimally sufficient driver combinations within the entire cohort. Under a proposed rule set, every sample is assigned to at most one rule in the rule set. Two rules are defined to be *family members* if one rule is a strict subset of the other. Rule sets cannot contain any two rules that are family members because it is a contradiction for both rules to be minimally sufficient.

CRSO is a stochastic optimization procedure over the space of possible rule sets. When a sample is assigned to a rule in a rule set RS, the events that comprise the rule are considered to be drivers within that sample. The *objective function score* for RS, J(RS), is defined to be the reduction in total statistical penalty under RS compared to the null rule set (Materials and Methods). Figure 1B shows an example of a rule set consisting of two rules applied to the miniature melanoma dataset. The total penalty is greatly reduced once samples are assigned to rules and the corresponding events are designated as drivers. The *coverage* of a rule set is the percentage of samples that can be assigned to at least one rule in the rule set. In Fig 1B, the coverage of RS is 80%.

The goal of CRSO is to find the rule set that achieves the best balance of J(RS), coverage and rule set size, which we call the core rule set. CRSO uses a four-phase procedure (Fig 1C) to first find the highest scoring rule set of size $K$ for $K \in \{1 \ldots 40\}$. The core rule set is then determined from among all of the solutions of size $K$. A subsampling process is used to identify an expanded list of generalized core rules (GCRs). A confidence score is determined for each GCR to be the frequency of inclusion in the subsampled iterations. The subset of GCRs that have confidence levels above 50 comprise the consensus GCRs (con-GCR), and by definition cannot contain family members. The full CRSO methodology is presented in Materials and Methods.

We applied CRSO to 19 cancer types obtained from TCGA (Table 1). The average number of events per cancer type was 86.4. There was a wide distribution of the number of candidate drivers per patient within individual cancers. In 17 out of 19 cancer types, the median number of total candidate drivers (SMGs plus SCNVs) was $\geq 6$, and in 6 cancer types, the median number of total

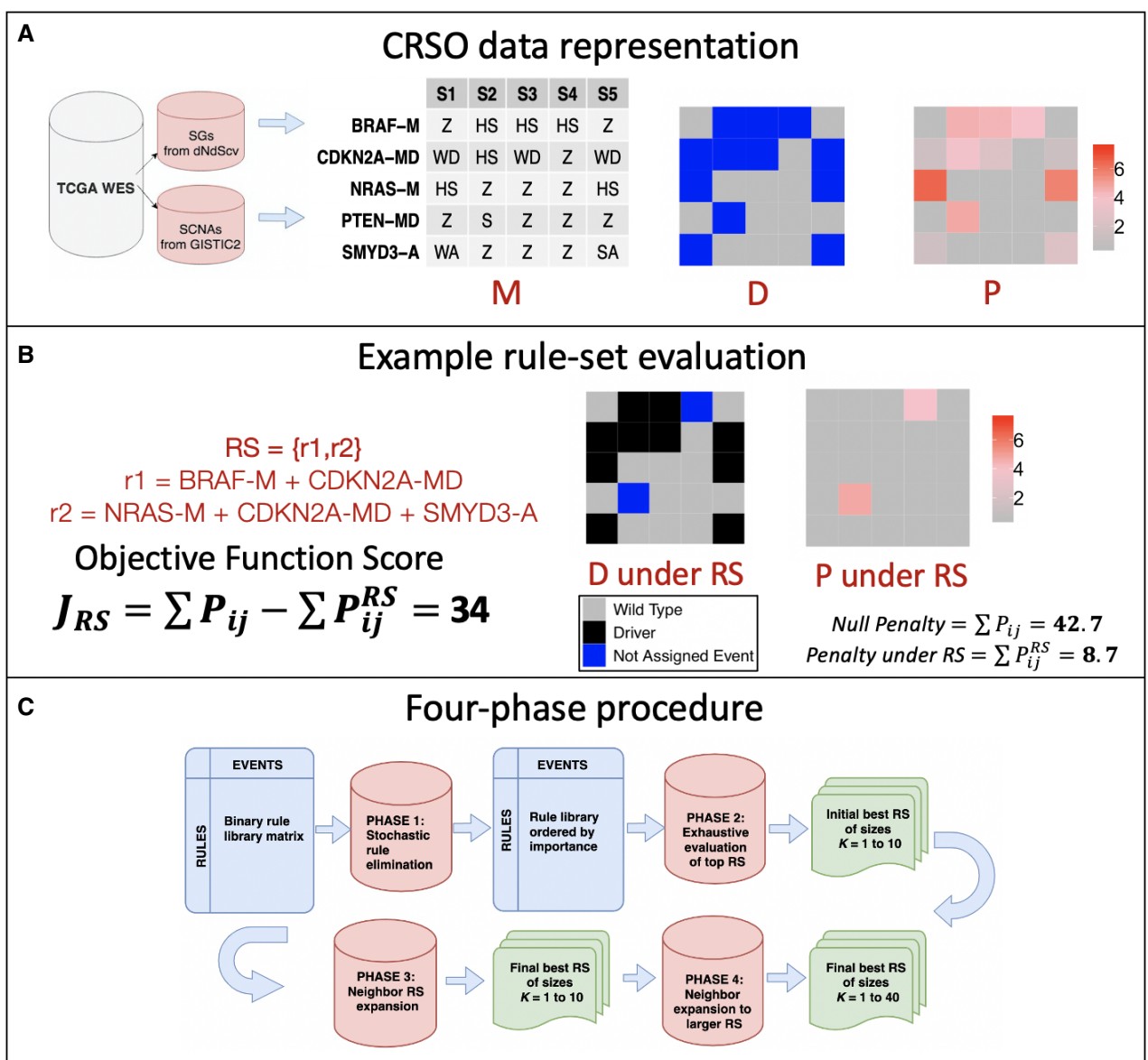

**Figure 1. CRSO at a glance.**

A CRSO representation of small dataset extracted from TCGA melanoma (SKCM). Candidate driver events identified by dNdScv and GISTIC2. Data are first represented as categorical matrix **M**, where each event can take one of several observation types, depending on the event type. Event types are indicated by the suffix, -M for mutations, -A for amplifications, and -MD for hybrid mutDels. Wild-type observations are represented as Z for all event types. Mutations are hotspot (HS), loss (L), splice site (S), or in-frame indel (I). Amplifications are either weak (WA) or strong (SA). Similarly, deletions are either weak (WD) or strong (SD). MutDel hybrid events can take values in any of the mutation or deletion observation types. Events in **M** are represented as a binary matrix **D** for making rule assignments (the square at position $ij$ is blue if sample $j$ harbors event $i$ and is gray if sample $j$ is wild type for event $i$). The penalty matrix **P** contains penalties for each possible observation-specific, patient-specific, event-specific passenger probabilities.

B Objective function calculated for an example rule set. Under the proposed rule set, the assigned events are designated as drivers, and the corresponding penalties are reduced to 0.

C Workflow of four-phase procedure for identifying the best rule sets of size $K$, for $K$ in 1 to 40.

candidate drivers was $\geq 10$ (Appendix Fig S1). CRSO identified an average of 11.1 core rules, covering an average of 70.3% of samples per cancer type. Of the 210 total core rules identified, 137 (65%), 47 (22%), 13 (6%), 12 (6%), and 1 (< 0.1%) consisted of 2, 3, 4, 5, and 6 events, respectively. The output of each CRSO run is a detailed report containing summaries and visual representations of the results (Datasets EV1–EV19).

**CRSO performance on simulated data**

In order to assess CRSO's ability to detect known driver combinations, we initially used simulations with known ground truth rule sets. Ten simulated datasets with ground truth rule sets of size $ntr$ were randomly generated for each $ntr \in \{2\ldots 20\}$, for a total of 190 simulations. Each dataset contained 100 events and 400 samples,

Table 1. Summary of 19 TCGA tissue types investigated with CRSO.

| Cancer | Full name | Samples | Events (M/C) | RCT | Rules | Kc | Cov | MRS | con-GCRs |
|--------|-----------|---------|--------------|-----|-------|-----|-----|-----|----------|
| BLCA | Bladder urothelial CA | 392 | 113 (47/66) | 0.033 | 1730 | 14 | 76 | 2.21 | 15 |
| BRCA | Breast invasive CA | 963 | 90 (28/62) | 0.03 | 449 | 11 | 58 | 2 | 13 |
| CESC | Cervical and endocervical cancers | 191 | 82 (23/59) | 0.031 | 173 | 12 | 53 | 2.08 | 15 |
| COAD | Colon AD | 362 | 97 (35/62) | 0.039 | 1792 | 14 | 81 | 3.64 | 10 |
| ESCA | Esophageal CA | 184 | 92 (12/80) | 0.054 | 1894 | 13 | 84 | 3.23 | 10 |
| GBM | Glioblastoma multiforme | 273 | 78 (15/63) | 0.033 | 186 | 11 | 77 | 2.27 | 10 |
| HNSC | Head and neck squamous cell CA | 505 | 94 (25/69) | 0.032 | 926 | 10 | 68 | 2.5 | 7 |
| KIRC | Kidney renal clear cell CA | 433 | 39 (13/26) | 0.014 | 66 | 10 | 52 | 2 | 11 |
| LGG | Brain lower-grade glioma | 513 | 67 (19/48) | 0.031 | 210 | 5 | 65 | 2.4 | 6 |
| LIHC | Liver hepatocellular CA | 366 | 75 (18/57) | 0.03 | 146 | 12 | 56 | 2 | 16 |
| LUAD | Lung AD | 478 | 93 (22/71) | 0.031 | 291 | 10 | 60 | 2.1 | 11 |
| LUSC | Lung squamous cell CA | 178 | 112 (37/75) | 0.045 | 1588 | 14 | 83 | 2.93 | 10 |
| OV | Ovarian serous cystAD | 455 | 77 (6/71) | 0.075 | 1990 | 12 | 86 | 3.08 | 12 |
| PAAD | Pancreatic AD | 126 | 64 (11/53) | 0.032 | 592 | 6 | 74 | 3 | 3 |
| PRAD | Prostate AD | 492 | 73 (13/60) | 0.03 | 301 | 10 | 51 | 2.5 | 6 |
| READ | Rectum AD | 120 | 67 (13/54) | 0.042 | 1260 | 12 | 89 | 3.17 | 9 |
| SKCM | Skin Cutaneous Melanoma | 290 | 71 (20/51) | 0.031 | 197 | 10 | 69 | 2 | 9 |
| STAD | Stomach AD | 391 | 121 (37/84) | 0.031 | 1690 | 13 | 64 | 2.46 | 14 |
| UCEC | Uterine Corpus Endometrial CA | 242 | 136 (39/97) | 0.037 | 1515 | 11 | 84 | 2.36 | 7 |

Full Name: AD short for adenocarcinoma, CA short for carcinoma. Events: total number of events as determined by dNdScv and GISTIC2, (M/C) is number of mutations and CNVs. RCT: rule coverage threshold, chosen to be 0.03 or the smallest threshold satisfied by at most 2,000 rules, except in the case of KIRC. Rules: number of rules in starting rule library. Kc: core rule set size. Cov: core rule set coverage. MRS: mean core rule set size. con-GCRs: number of consensus generalized core rules.

chosen to approximate the mean number of events and samples across the TCGA datasets. Passenger rates for events and samples were sampled from a pooled distribution of real data passenger rates (Materials and Methods, Appendix Fig S2). We evaluated CRSO using 3 metrics: sensitivity, precision, and rule assignment accuracy. Sensitivity was calculated to be the percentage of ground truth rules included in the consensus GCRs (con-GCRs). Precision was calculated to be the percentage of con-GCRs that are in the ground truth rule set. Assignment accuracy was the percentage of samples identically assigned in the core RS assignment and the ground truth assignment.

The mean sensitivity was at least 0.85 for all $ntr \in \{2\dots20\}$ and was $\geq 0.90$ for 16/19 $ntr$ values (Appendix Table S3, Fig 2A). The mean precision was 0.73 for $ntr = 2$ and was $\geq 0.96$ for the other 18 true rule set sizes (Appendix Table S3, Fig 2B). The lowest mean accuracy was observed for $ntr = 2$, at 0.76, and was above 0.80 for all other $ntr$ values. Starting with $ntr = 3$, for which the mean sensitivity was 1.0 and the mean accuracy was 0.98, the general trend was that sensitivity and accuracy decreased as $ntr$ increased. For $ntr = 2$, 3 simulations performed poorly with accuracies below 0.4 and the other 7 simulations all had accuracies greater than 0.90 (Appendix Fig S3), suggesting that CRSO may be susceptible to over-fitting when the number of true rules is only 2.

Across 190 simulated datasets, CRSO was able to navigate an intractable search space of possible rule sets (e.g., for a library of 1,500 rules, there are $> 10^{45}$ possible rule sets of size $\leq 20$) and identify ground truth rules comprising between 2 and 6 events with high precision and high sensitivity. We compared CRSO's performance on the simulated datasets with that of pairwise co-occurrence tests—a popular strategy for detecting driver-gene cooperation. For each simulation, the Fisher's exact test was performed for all pairs of events, and the detected co-occurrences were compared to the ground truth duos, i.e., the superset of all pairs of events that co-occur within at least one ground truth rule. Events with 0 or 1 occurrence were excluded. Multiple-hypothesis correction for each simulation experiment was performed using a false discovery rate of 5%. Sensitivities were calculated as the percentage of true pairwise interactions that were identified as statistically co-occurrent. Precisions were calculated as the percentage of identified pairwise interactions that were part of at least one ground truth rule.

The mean sensitivity achieved by the Fisher's test was $0.53 \pm 0.18$ over all $ntr$ and ranged from a high of 0.88 for $ntr = 3$ to a low of 0.30 for $ntr = 20$ (Fig 2A). The mean precision achieved by the Fisher's test was $0.41 \pm 0.09$ over all $ntr$ and ranged from a low of 0.29 for $ntr = 2$ to a high of 0.60 for $ntr = 19$ (Fig 2B). The mean sensitivities and precisions achieved by CRSO outperformed those of the Fisher's test for all $ntr$, with an average sensitivity improvement of 0.4 and an average precision improvement of 0.57 (Fig 2).

To better understand the limitations of pairwise co-occurrence tests, we applied the Fisher's exact test to a version of the simulations having uniform passenger rates across samples and events, and to a version of the simulations with zero passenger events. Appendix Fig S2 shows an example simulation with zero passenger events (**Dz**), uniformly distributed passenger events (**Du**), and

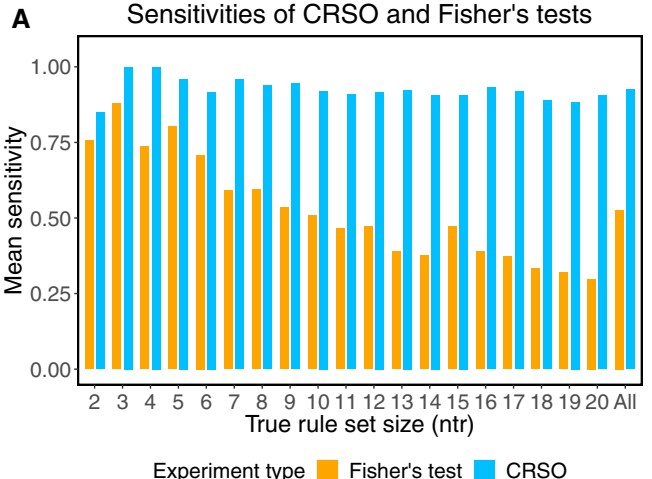

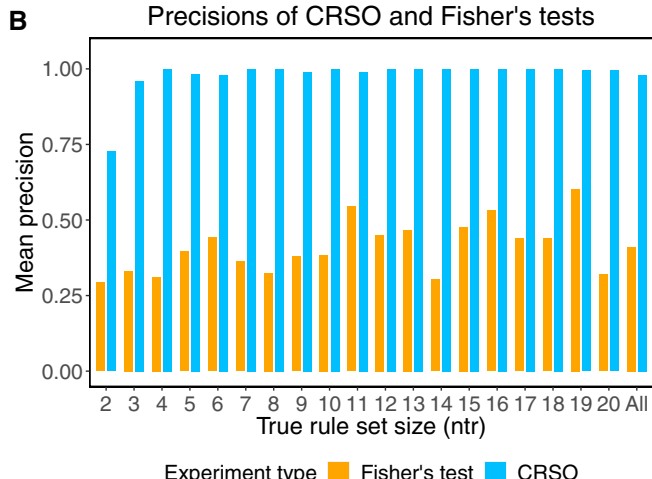

**Figure 2. Performance of CRSO and Fisher's tests applied to 190 ground truth simulations.**

Results are grouped according to ground truth rule set size (*NTR*). The group denoted "All" is the global mean over all *NTR*.
A  Mean sensitivities of CRSO predictions evaluated against ground truth rules (light blue bars) and Fisher's test predictions evaluated against ground truth duos (orange bars).
B  Mean precisions of CRSO predictions evaluated against ground truth rules (light blue bars) and Fisher's predictions evaluated against ground truth duos (orange bars).

realistically distributed noise rates that were used for the CRSO performance evaluation (**Dr**). The mean sensitivity for **Du** was $0.47 \pm 0.20$ across all *ntr* and was comparable to the mean sensitivity for **Dr** (Appendix Fig S5A). The mean precision for **Du** was $0.96 \pm 0.03$ across all *ntr* and was far superior to the those for **Dr** (Appendix Fig S5B). The precision results suggest that unaccounted-for heterogeneity in passenger rates across samples and events lead to an excess of false positives, consistent with the findings of Canisius *et al* (2016). The precision was always 1.0 for **Dz**, since it is impossible to have false positives in the absence of noise. The mean sensitivity for **Dz** was $0.76 \pm 0.11$ across all *ntr* and ranged from a high of 1.0 for *ntr* = 2 to a low of 0.64 for *ntr* = 20 (Fig 2A). The sensitivities for simulations without any noise show that many false negatives result directly from the ground truth rule set structure.

**Application to TCGA melanoma data**

We next present results from 290 TCGA cutaneous melanomas (SKCM, Fig 3). We use this as an exemplar dataset as SKCM is a heterogeneous cancer type with some known driver combinations (Gen, 2004; Dankort *et al*, 2009; Griffin *et al*, 2017; Stramucci *et al*, 2018). We used as input, 20 SMGs from dNdScv, 34 SCNV deletions and 20 SCNV amplifications from GISTIC2 narrow peak calls. Three genes, *CDKN2A, PTEN,* and *B2M,* were represented as hybrid "mutDel" events as they were both significantly mutated and deleted in the dataset (see Materials and Methods). Figure 3 shows the frequencies (Fig 3A) and passenger penalties (Fig 3B) of the 25 most frequent events.

The candidate rule library contained 197 rules that satisfied the coverage threshold of 3%, of which 165 consisted of two events and 32 consisted of three events. The core rule set was determined to be the best rule set of size $K = 10$, as this was the smallest rule set that exceeded both the performance and coverage thresholds of at least 90% (Appendix Fig S6A and B). There were no valid rule sets that

satisfied the minimum samples assigned (*msa*) threshold of 9 samples for $K \geq 18$. Figure 4 shows the optimal assignment under the core RS and the corresponding reduction in penalty (Fig 4B and C). Thirty percent of the SKCM patients do not satisfy any of the core rules, indicated by the gray color bar in Fig 4A.

Generalized core (GC) analysis using 100 subsampling iterations identified 9 con-GCRs (Materials and Methods). Although all of the melanoma con-GCRs contained 2 events, comparison of the GCRs with GC duos reveals that some con-GCRs appear as part of larger rules in some GC iterations (Appendix Fig S7). For example, *BRAF-M + CDKN2A-MD* and *BRAF-M + PTEN-MD* are observed as core rules in 83 and 84% of GC iterations, respectively, but are both observed as core duos in 100% of GC iterations. The difference in GCD and GCR confidence scores indicates that CRSO is 100% confident that *BRAF-M + PTEN-MD* and *BRAF-M + CDKN2A-MD* are both essential combinations in a subset of melanoma patients, but is approximately 84% confident that these duos are independently sufficient to produce melanomas.

The core RS and con-GCRs can be considered complementary best rule sets, with the core RS providing the single best performing RS and corresponding assignment over the full dataset, and the con-GCRs providing a robust set of rules with quantified confidence scores. The union of the melanoma core RS and con-GCRs comprise 11 distinct rules and are dominated by rules that contain either *BRAF* or *NRAS* mutations (Table 2). Of the 11 rules, 5 contain *BRAF*, 5 contain *NRAS,* and only 1 rule, *B2M-MD + FMN1/SNORD77-D*, contains neither. Hotspot mutations in *BRAF* and *NRAS* define the two major subtypes of melanoma that are mutually exclusive, with 50% of patients harboring *BRAFV600E* mutations and 30% of patients harboring *NRAS* hotspot mutations (Cancer Genome Atlas Network, 2015a). CRSO prioritized rules containing these events because it is improbable that these highly recurrent hotspot mutations would have happened by chance.

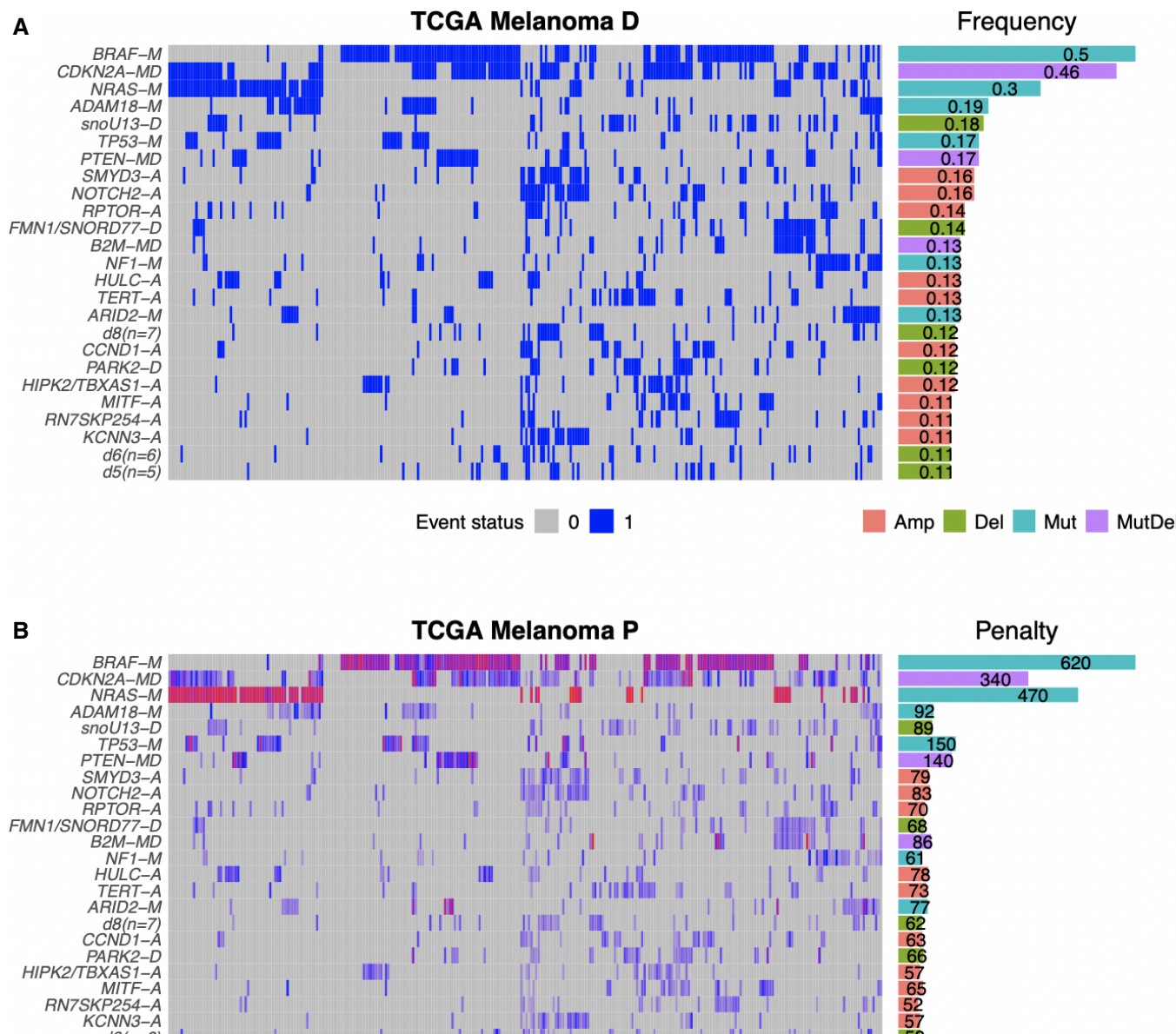

**Figure 3. CRSO representation of TCGA melanoma dataset.**

A  Binary representation of top-25 most frequent events. Horizontal bars on the right show event frequencies across the population.
B  Penalty matrix for top-25 most frequent events. Horizontal bars on the right show total event penalties.

Data information: Event types indicated by suffixes: -A for amplifications, -D for deletions, -M for mutations, and -MD for mutDel hybrid events.

## Expected melanoma combinations detected by CRSO

We categorized the 11 rules as either expected (5 rules) or potentially novel (6 rules) based on literature review. The 5 expected rules involve combinations of either *BRAF* or *NRAS* with well-studied tumor suppressors: (i) *NRAS-M + TP53-M*, (ii) *BRAF-M + TP53-M*, (iii) *NRAS-M + CDKN2A-MD*, (iv) *BRAF-M + CDKN2A-MD,* and

(v) *BRAF-M + PTEN-MD*. The first 4 of these rules are instances of co-occurring MAPK3 pathway activation with P53 inactivation—a synergy that is known to promote carcinogenesis (Gen, 2004; Stramucci *et al,* 2018). Evidence in multiple cancer types supports the cooperation between activating mutations in *KRAS* and loss of the G1/S checkpoint by inactivation of *CDKN2A* or *TP53* (Aguirre *et al,* 2003; Courtois-Cox *et al,* 2008; Schuster *et al,* 2014). CRSO

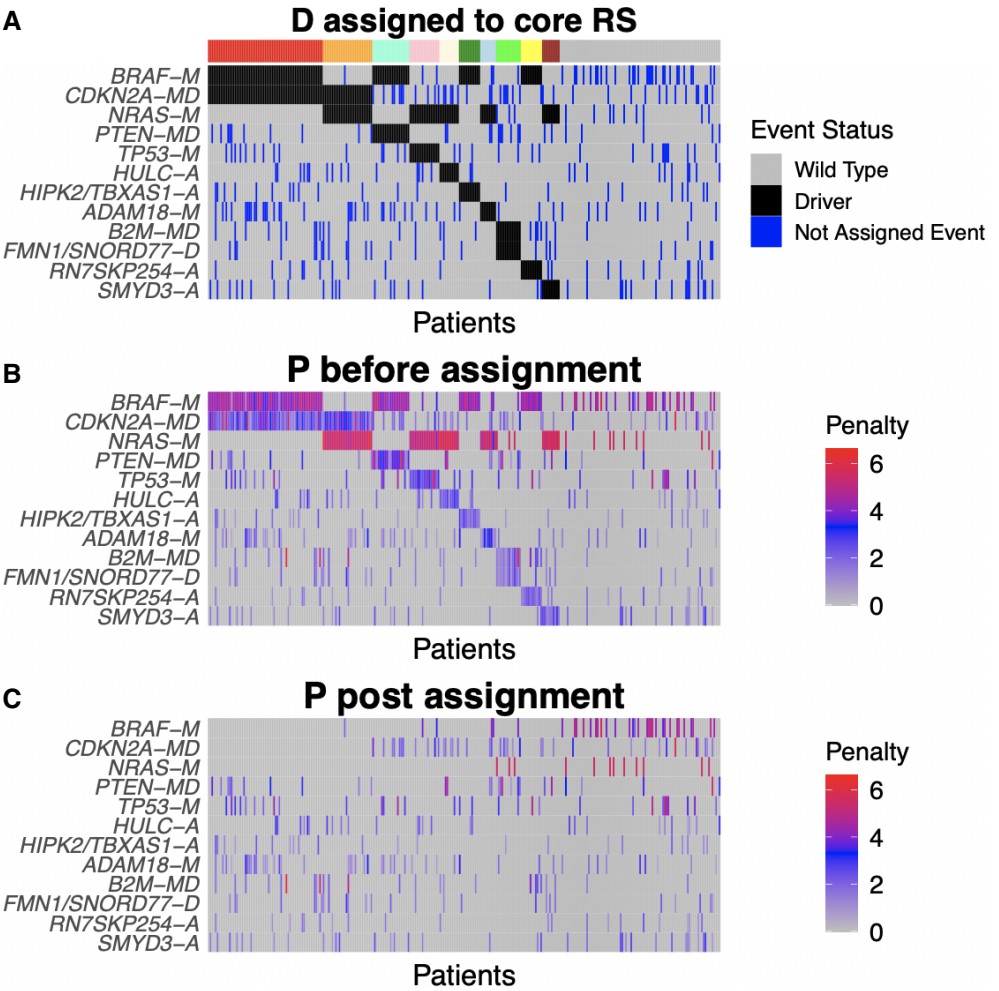

**Figure 4. Melanoma core rule set assignment.**

A Heatmap of the binary alteration matrix **D** under core rule set assignment. Events are ordered by frequency. Samples are ordered according to rule set membership, as indicated by the color bar. The right-most group (gray bar) are not assigned to any rule. For each sample, assigned events are shown in black, and unassigned events are shown in blue.

B, C Heatmaps of the passenger penalty matrix **P** before and after assignment to the core rule set.

Data information: Event types indicated by suffixes: -*A* for amplifications, -*D* for deletions, -*M* for mutations, and -*MD* for mutDel hybrid events.

identified *KRAS-M* + *TP53-M* as a consensus GCR in 3 other cancer types, LUAD, PAAD, and STAD, and identified *KRAS-M* + *CDKN2A-MD* as a con-GCR in LUAD and PAAD. Co-occurrence of *BRAFV600E* and *CDKN2A* loss defines a subset of pediatric brain tumors that are responsive to combined treatment with *BRAF* and *CDK4/6* inhibitors (Huillard *et al*, 2012). Unlike *CDKN2A* and *TP53*, *PTEN* was inferred to exclusively cooperate with *BRAF*. Supporting the minimal sufficiency of *BRAF-M* + *PTEN-MD*, co-occurrence of *BRAF600E* and *PTEN* loss was shown to induce metastatic melanoma in mouse models, while *BRAFV600E* alone only produced benign nevi in the mice (Dankort *et al*, 2009). *PTEN* loss is a mechanism of primary and acquired resistance to *BRAF* inhibitors, inspiring the development of combination strategies co-targeting the *BRAF/MEK* and *PI3K/AKT* pathways in tumors harboring both *BRAFV600E* and *PTEN* loss (Fedorenko *et al*, 2011; Manzano *et al*, 2016; Griffin *et al*, 2017).

These 5 rules exemplify CRSO's ability to identify combinations with experimental evidence of being minimally sufficient to drive

cancer, and with established utility as biomarkers and sources of rational combination strategy. Although these combinations are not novel, CRSO prioritized these rules without any prior biological knowledge. By contrast, 2 published methods for systematically identifying driver combinations were applied to the same TCGA melanoma dataset and did not identify any of these 5 combinations (Mina *et al*, 2017; Al Hajri *et al*, 2020). Since most strategies are based on pairwise co-occurrence, the omission of these well-studied synergies in melanoma further demonstrates the inappropriateness of statistical co-occurrence as a criterion for detecting driver combinations.

**Novel melanoma combinations detected by CRSO**

The other 6 rules in Table 2 involve lesser-known drivers and may represent novel biological subtypes of melanoma. *NRAS-M* + *HULC-A* was prioritized as both a con-GCR (conf. = 71) and as part of the core RS. *HULC* is a long non-coding RNA that has been identified as

Table 2. Melanoma core RS and consensus generalized core rules (con-GCRs).

| Rule | Core Type | P1-rank | Conf | Coverage | SJ | % Assigned |
|---|---|---|---|---|---|---|
| NRAS-M + TP53-M | Both | 4 | 100 | 6.6% ($r = 29$) | 168 ($r = 8$) | 89 |
| BRAF-M + RN7SKP254-A | Both | 11 | 98 | 6.6% ($r = 32$) | 137 ($r = 21$) | 63 |
| BRAF-M + HIPK2/TBXAS1-A | Both | 6 | 97 | 7.9% ($r = 13$) | 161($r = 10$) | 52 |
| CDKN2A-MD + NRAS-M | Both | 2 | 84 | 14% ($r = 2$) | 327 ($r = 2$) | 68 |
| BRAF-M + PTEN-MD | Both | 3 | 84 | 11% ($r = 3$) | 227 ($r = 3$) | 68 |
| BRAF-M + CDKN2A-MD | Both | 1 | 83 | 26% ($r = 1$) | 524 ($r = 1$) | 86 |
| HULC-A + NRAS-M | Both | 5 | 71 | 6.6% ($r = 28$) | 167 ($r = 9$) | 58 |
| B2M-MD + FMN1/SNORD77-D | Both | 10 | 66 | 9% ($r = 6$) | 144 ($r = 36$) | 54 |
| ADAM18-M + NRAS-M | Core | 9 | 48 | 6.9% ($r = 20$) | 146 ($r = 11$) | 45 |
| NRAS-M + SMYD3-A | Core | 15 | 40 | 5.5% ($r = 40$) | 131 ($r = 20$) | 62 |
| BRAF-M + TP53-M | Con-GCR | 8 | 51 | 8.3% ($r = 9$) | 156 ($r = 7$) | – |

Core Type: Core RS, con-GCR, or both. P1-rank is the phase 1 ranking of rule. Confidence (conf) is generalized core rules confidence level, con-GCRs defined as rules with confidence above 50. Coverage is the percentage of samples that cover the rule (and ranking among all rules in the rule library). SJ is the single-rule performance (and ranking). % Assigned is the percentage of samples that satisfy a rule are assigned to the rule (applicable only to core rules).

a driver of tumorigenesis in multiple cancers (Yu *et al*, 2017; Klec *et al*, 2019; Ghafouri-Fard *et al*, 2020), including liver cancer, osteosarcoma, cervical cancer, pancreatic, stomach, and ovarian cancers (Panzitt *et al*, 2007; Zhang *et al*, 2016; Chen *et al*, 2017a; Wang *et al*, 2020; Li *et al*, 2020; Lu *et al*, 2020). *HULC* expression is a predictor of poor prognosis across multiple cancers (Peng *et al*, 2014; Jin *et al*, 2016; Fan *et al*, 2017; Chen *et al*, 2017b), and *HULC* silencing enhanced the effectiveness of chemotherapy in stomach cancer cell lines (Zhang *et al*, 2016). The involvement of *HULC* in the tumorigenesis of a subset of *NRAS*-mutant melanomas appears to be unreported and merits further investigation as a possible novel discovery with therapeutic implications.

*NRAS-M + SMYD3-A* (conf. = 40) was identified as part of the CRSO core RS. *SMYD3* is a member of the histone lysine methyltransferases enzyme family, and upregulates transcription of a plethora of oncogenes in multiple cancer types, including *CDK2* and *MMP2* in hepatocellular carcinomas (Wang *et al*, 2019), *BCLAF1* in bladder cancer (Shen *et al*, 2016), androgen receptors in prostate cancer (Liu *et al*, 2013), *EGFR* in renal cell carcinoma (Liu *et al*, 2020a), and *MYC* and *CTNNB1* in colon and liver cancers (Sarris *et al*, 2016). Mazur *et al* (2014) showed that *SMYD3* methylation of *MAP3K2* leads to upregulation of MAP kinase signaling and promotes carcinogenesis in *RAS* mutated lung and pancreatic cancers. The authors further showed that preventing *SMYD3* catalytic activity in mouse models with oncogenic *RAS* mutations inhibited tumor development (Mazur *et al*, 2014). This presents a direct biological link between *SMYD3* expression and *RAS* mutations, and nominates *SMYD3* as a drug target for *RAS*-driven cancers. As with *HULC*, the possible role of *SMYD3* in melanoma appears to be unreported.

CRSO identified two synergies exclusive to *BRAF-M* with very high confidence: *BRAF-M + HIPK2/TBXAS1-A* (conf. = 97) and *BRAF-M + RN7SKP254-A* (conf. = 98). Although we annotated amplifications according to the GISTIC2 narrow peak regions, these two amplification events are part of large wide peak regions containing many genes. The wide peak annotated by *HIPK2* and *TBXAS1* contains 72 genes, including *BRAF*. *BRAF* amplification is a

mechanism of acquired resistance to *BRAF* inhibitors in *BRAFV600E* melanomas (Corcoran *et al*, 2010; Villanueva *et al*, 2013; Nathanson *et al*, 2013; Stagni *et al*, 2018). Whereas overall patients harboring *BRAF-M* had longer progression-free intervals (PFIs) than *BRAF* WT patients (Fig 5A), those patients harboring both *BRAF-M* and *HIPK2/TBXAS1-A* had shorter PFI compared to patients harboring *BRAF-M* without *HIPK2/TBXAS1-A* (Fig 5B, Cox proportional hazards $P = 0.016$, $P_{Adj} = 0.084$, explained below in subsection "Associations of rules with patient outcomes"). Because TCGA tumors are primary and treatment-naive, this observation suggests that *BRAF* amplification and *BRAFV600E* may be a sufficient combination for tumor formation, and a cause of intrinsic resistance to *BRAF* inhibition.

The wide amplification peak annotated by *RN7SKP254* is on chromosome 15q26.2 and contains 169 genes. Four genes were classified as cancer genes based on the Sanger Institute Cancer Gene Census (Sondka *et al*, 2018): *BLM*, *IDH2*, *NTRK2*, and *CRTC3*. This rule was prioritized by CRSO as the second highest confidence rule (98%) despite ranking 32nd in coverage (6.6%) and 21st in SJ ranking, suggesting that this amplification may merit further investigation.

### Known and novel findings in TCGA

In this section, we highlight some findings from the TCGA CRSO results. Appendix Table S6 presents all con-GCRs and provides a concise snapshot of the results from all cancer types.

### Known 3-gene combinations in brain and colorectal cancers

One of the distinguishing features of CRSO is the ability to identify functionally relevant co-occurrences involving 3 or more events. The rule *ATRX-M + IDH1-M + TP53-M* was identified as a con-GCR in both LGG (conf.: 71) and GBM (conf.: 63), despite only occurring in 3.7% of GBM samples. This 3-gene combination defines a well-known subtype of brain cancers with differential prognosis, and has been experimentally shown to induce carcinogenesis (Jiao *et al*, 2012; Cancer Genome Atlas Network, 2015d; Modrek *et al*, 2017). Similarly, the well-studied combination of *APC-M + KRAS-M + TP53-M*

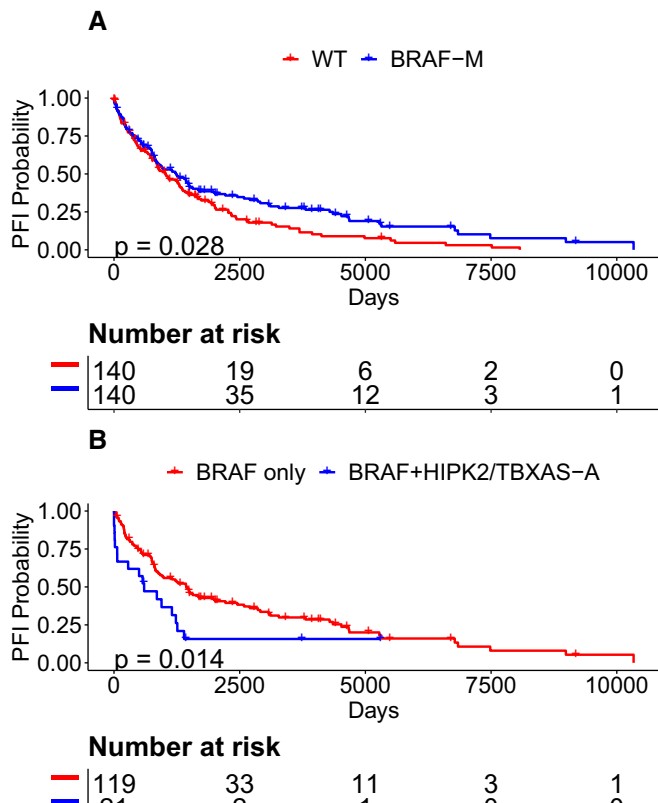

**Figure 5.** *BRAF* melanomas with *HIPK2/TBXAS* amplification have poor PFI.

*HIPK2/TBXAS1-A* annotates an amplified region that contains *BRAF* and defines a subtype of *BRAF* mutant SKCM patients with poor PFI.

A  *BRAF* patients have improved PFI compared to non-*BRAF* patients.

B  SKCM patients with *BRAF* + *HIPK2/TBXAS1-A* have worse PFI than patients with only *BRAF*.

Data information: *P* values calculated from Kaplan–Meier estimator.

in colorectal cancers (Fearon & Vogelstein, 1990; Cho & Vogelstein, 1992; Smith *et al*, 2002) was identified as a con-GCR in READ (conf.: 79) and as a non-consensus GCR in COAD (conf.: 42).

### CRSO prioritized *NFE2L2* combinations in multiple cancers

*NFE2L2* mutations were identified as part of a con-GCR in 4 cancer types: LUSC, HNSC, ESCA, and BLCA. All of the con-GCRs involving *NFE2L2* ranked very low in terms of single-rule metrics (Table 3), suggesting that *NFE2L2* is accounting for samples not accounted for by any higher-ranking rules. *NFE2L2* encodes NRF2, a key transcription factor that regulates cellular response to oxidative stress (Rojo de la Vega *et al*, 2018). Initially, NRF2 activation was identified as a mechanism of cellular protection against cancer (Zhang, 2006; Li & Kong, 2009). However, we now have evidence that constitutive NRF2 activation via mutations in *KEAP1* or recurrent *NFE2L2* exon 2 deletions can drive tumor proliferation and metastasis (Goldstein *et al*, 2016; Rojo de la Vega *et al*, 2018). *NFE2L2* mutations have recently been shown to define subtypes with differential prognosis in lung and head and neck cancers (Frank

*et al*, 2018; Namani *et al*, 2018; Xu *et al*, 2020; Liu *et al*, 2020b). Several upcoming clinical trials are designed to explore the therapeutic benefit of compounds that inhibit NRF2 in advanced cancer patients harboring mutations in *NFE2L2* or *KEAP1* (ClinicalTrials.gov, NCT02417701; ClinicalTrials.gov, NCT04267913; ClinicalTrials.gov, NCT03872427). The rules containing *NFE2L2* may help identify subsets of patients that are sensitive to *NFE2L2* inhibition.

### CRSO prioritized a rare combination in head and neck cancers

The rule *CASP8-M* + *HRAS-M* was identified in HNSC as the third highest ranking GCR (conf. = 73) even though it ranks 591st in coverage and 301st in SJ out of 926 rules. Patients that have these two mutations and are also wild type for *TP53* have been shown to define a biologically distinct subtype characterized by low SCNV burden (Cancer Genome Atlas Network, 2015c). The prioritization of *CASP8-M* + *HRAS-M* despite low coverage (3.8%) demonstrates CRSO's capability to systematically identify non-obvious, biologically meaningful combinations among the myriad of possible combinations.

### CRSO prioritized rules containing *ALB* mutations in liver cancer

Two high confidence GCRs were identified in LIHC involving *ALB* mutations: *ALB-M* + *CTNNB1-M* (conf. = 98) and *ALB-M* + *TP53-M* (conf. = 80). *ALB* was experimentally shown to be a tumor suppressor in hepatocellular carcinomas (Nojiri & Joh, 2014), and has been discussed as an important part of the somatic mutation landscape (Cancer Genome Atlas Network, 2017). Cooperations involving *ALB* have not been systematically reported, and the two combinations we identified may help inform context-dependent treatments of *ALB* mutant patients. Evidence of the relevance of these rules on PFI is presented below in "Associations of rules with patient outcomes".

### CRSO identified a novel 3-gene combination in bladder cancer

*ARID1A-MD* + *SOX4-A* + *TP53-M* was identified as a con-GCR in BLCA (conf. = 79, coverage = 7%). *SOX4* over-expression has been studied experimentally and has been reported to be an important contributor in bladder cancer tumorigenesis (Shen *et al*, 2015; Moran *et al*, 2019). The hypothesis that *SOX4* cooperates with *ARID1A* and *TP53* to initiate bladder carcinogenesis is novel and may merit further experimental investigation.

### Associations of rules with patient outcomes

We evaluated whether stratifying patients according to con-GCRs predicted by CRSO instead of individual driver events provided extra prognostic information. We restricted this analysis to events that appear in more than one con-GCR (referred to as multi-rule events), since these events are predicted by CRSO to occur in distinct genetic contexts. For every con-GCR, R, containing a multi-rule event, E, we applied a univariate Cox proportional hazards (Cox-PH) analysis to compare the progression-free intervals (PFIs) of samples that satisfy R versus samples that harbor E but do not satisfy R. PFI data were obtained from a recently published resource for TCGA outcome analysis (Liu *et al*, 2018), and the use of PFI

**Table 3.** Consensus GCRs involving *NFE2L2*.

| Tissue | Rule | Confidence | Coverage | SJ | NE |
|--------|------|-----------|----------|-----|-----|
| LUSC | *NFE2L2-MA + CSMD3-M + SOX2-A* | 79 (r = 3) | 8.4% (r = 230) | 168 (r = 8) | 3 |
| HNSC | *NFE2L2-MA + CDKN2A-MD + TP53-M* | 50 (r = 7) | 8.7% (r = 77) | 137 (r = 21) | 3 |
| ESCA | *NFE2L2-MA + ACTRT/MYNN-A + TP53-M* | 59 (r = 9) | 6.0% (r = 1280) | 161(r = 10) | 3 |
| BLCA | *NFE2L2-MA + CDKN2A-MD* | 51 (r = 15) | 5.6% (r = 337) | 327 (r = 2) | 2 |

SJ is single-rule objective function score. NE is number of events. Ranking (r) is shown in parentheses.

as primary endpoint is consistent with the authors' recommendations for best practices. The result of each test is summarized with a $Z$ score, as recommended in Smith and Sheltzer (2018). Positive $Z$ scores indicate better outcomes in the cohort satisfying R versus the cohort that only harbors E, and negative $Z$ scores indicate the opposite.

For each cancer type, each multi-rule event, E, was tested against all of the GCRs that include E, provided that both the rule and event classes contained at least 10 patients. A total of 289 tests were performed across the 19 cancer types, and 21 (7.3%) associations were identified as having a Cox-PH $|Z| \geq 1.96$. Standard multiple-hypothesis correction procedures such as Bonferroni correction are not appropriate because the comparisons within a given cancer type are not independent. Instead, a permutation test was performed in order to address the multiple hypotheses associated with each multi-event rule. For each multi-rule event, the PFIs of the samples that contain the event were scrambled 1,000 times, and the smallest $P$ value attained by any of the rule versus event comparisons was stored for each iteration. An adjusted $P$ value,

$P_{Adj}$, was defined to be the fraction of iterations that have permuted $P$ values smaller than the $P$ value calculated from the real data. We identified 17 associations for which both Cox-PH $|Z| \geq 1.96$ and $P_{Adj} \leq 0.15$ (Table 4).

**CRSO combinations refine the classification of *IDH1*-mutant LGGs**

*IDH1* mutations occur in 78% of LGGs and are a biomarker for improved outcome in LGG patients (Cancer Genome Atlas Network, 2015d) (Fig 6A). Differential outcomes were detected between the event/rule pairings *IDH1-M* versus *IDH1-M + IC-M* and *IDH1-M* versus *IDH1-M + PIK3CA-M* that could further stratify *IDH1* mutant samples (Fig 6). Among the patients with *IDH1-M* (n = 397), those that also have *CIC-M* (n = 99) have better PFI than those that are *CIC* wild type (n = 298, Fig 6B, Cox-PH $Z = 2.5$, $P_{Adj} = 0.047$). On the other hand, the 31 *IDH1*-mutant samples that also harbor *PIK3CA-M* have worse PFI than those that are *PIK3CA* wild type (n = 366, Fig 6C, Cox-PH $Z = -2.1$, $P_{Adj} = 0.139$). Collectively, these results suggest that LGG patients could be stratified into 4

**Table 4.** Significant PFI associations among consensus GCRs.

| Tissue | Event | Rule | NR | NE | Z | P | $P_{Adj}$ |
|--------|-------|------|-----|-----|-----|-----|-----|
| BLCA | *ARID1A-MD* | *ARID1A-MD + SOX4-A + TP53-M* | 28 | 100 | 2.986 | 0.0028 | 0.006 |
| BLCA | *FGFR3-MA* | *CDKN2A-MD + FGFR3-MA* | 44 | 41 | −2.536 | 0.0112 | 0.021 |
| BLCA | *TP53-M* | *ARID1A-MD + SOX4-A + TP53-M* | 28 | 167 | 2.457 | 0.014 | 0.083 |
| BRCA | *PIK3CA-MA* | *CCND1/ORAOV1-A + PIK3CA-MA* | 109 | 298 | −2.167 | 0.0302 | 0.077 |
| BRCA | *TP53-M* | *PIK3CA-MA + TP53-M* | 130 | 163 | 2.574 | 0.0101 | 0.043 |
| KIRC | *RNA5SP200-A* | *BAP1-M + RNA5SP200-A* | 12 | 96 | −3.179 | 0.0015 | 0.005 |
| LGG | *IDH1-M* | *CIC-M + IDH1-M* | 99 | 298 | 2.499 | 0.0124 | 0.047 |
| LGG | *IDH1-M* | *IDH1-M + PIK3CA-M* | 31 | 366 | −2.1 | 0.0357 | 0.139 |
| LGG | *TP53-M* | *ATRX-MD + IDH1-M + TP53-M* | 191 | 59 | 2.12 | 0.034 | 0.055 |
| LIHC | *ARID1A-MD* | *ARID1A-MD + CTNNB1-M* | 23 | 61 | −2.886 | 0.0039 | 0.006 |
| LIHC | *CTNNB1-M* | *ARID1A-MD + CTNNB1-M* | 23 | 74 | −2.27 | 0.0232 | 0.079 |
| LIHC | *TP53-MD* | *ALB-M + TP53-MD* | 13 | 111 | −2.351 | 0.0187 | 0.150 |
| OV | *d2 (n = 15)* | *d2 (n = 15) + 4 Events* | 59 | 121 | −2.255 | 0.0242 | 0.062 |
| SKCM | *BRAF-M* | *BRAF-M + HIPK2/TBXAS1-A* | 21 | 119 | −2.407 | 0.0161 | 0.084 |
| UCEC | *PIK3CA-M* | *PIK3CA-M + PTEN-MD* | 89 | 38 | 2.221 | 0.0264 | 0.056 |
| UCEC | *PIK3CA-M* | *CTNNB1-M + PIK3CA-M* | 40 | 87 | 1.984 | 0.0472 | 0.110 |
| UCEC | *PIK3CA-M* | *PIK3CA-M + TP53-M* | 32 | 95 | −3.751 | 0.0002 | 0.000 |

Each experiment consisted of comparing the samples satisfying the rule with the samples harboring the event, but not satisfying the rule. NR and NE are number of samples in the rule and event classes, respectively. Z scores (Z) and p values (P) were calculated using univariate Cox-PH. Positive Z score indicates that the patients satisfying the rule have better prognosis than those who only satisfy the event. $P_{Adj}$ is adjusted p value based on a permutation test (described in text). The rule "d2(n = 15) + 4 Events" is short for "d2(n = 15) + *FKSG52/PDE4D-D + MECOM-A + MYC-A + TP53-M*".

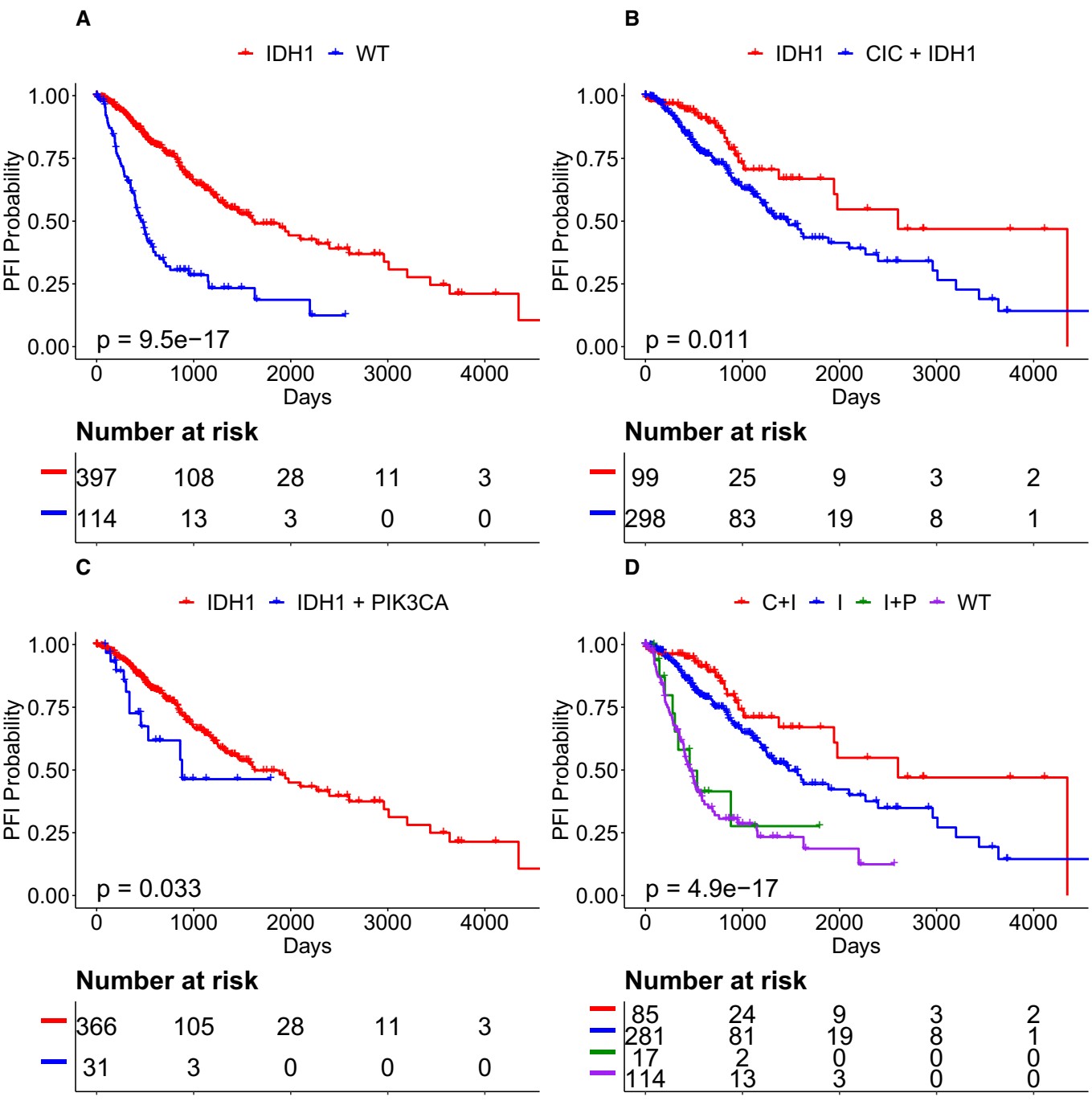

**Figure 6. CRSO rules refine the stratification of *IDH1*-mutant lower-grade gliomas (LGG).**

A *IDH1* mutations define a subset of LGG patients with better PFI.

B Patients with *IDH1* and *CIC* have better PFI than patients with *IDH1* only.

C Patients with *IDH1* and *PIK3CA* have worse PFI than patients with *IDH1* only.

D LGG patients can be stratified into four classes: *IDH1* wild type (WT), *IDH1* and *CIC* (C + I), *IDH1* + *PIK3CA* (I + P), and *IDH1* mutant samples that are wild type for both *PIK3CA* and *CIC* (I). Samples harboring *IDH1*, *PIK3CA*, and *CIC* were excluded from panel D (n = 14).

Data information: P values calculated from Kaplan–Meier estimator.

groups: *IDH1* wild type, *IDH1* mutant + *CIC* mutant, *IDH1* mutant + *PIK3CA* mutant, and *IDH1* mutant + *CIC/PIK3CA* wild type. *PIK3CA-M* appears to nullify the improvement in PFI conferred by *IDH1*, and

that *CIC-M* enhances the improvement in PFI (Fig 6D). The improved prognosis of *IDH1-M* + *CIC-M* relative to other *IDH1* mutant LGGs has been previously reported and has been

independently characterized by p1/q19 co-deletions, which overlap highly with *CIC* mutations (Jiao *et al*, 2012; Brat *et al*, 2015; Cancer Genome Atlas Network, 2015d). We did not find literature evidence of the poor prognosis of *IDH1* tumors harboring *PIK3CA* mutations in LGGs.

### CRSO identifies potential multi-gene biomarkers in liver cancer

LIHC patients with both *ARID1A-MD* and *CTNNB1-M* (n = 23) had worse PFI expectancy than patients with *ARID1A-MD* but not *CTNNB1-M* (Table 4, $n = 61$, Cox-PH Z = –2.89, $P_{Adj} = 0.006$). The patients satisfying this rule also had worse PFI expectancy than those with *CTNNB1-M* but not *ARID1A-MD* ($n = 74$, Cox-PH $Z = -2.27$, $P_{Adj} = 0.079$). Surprisingly, neither *ARID1A-MD* nor *CTNNB1-M* is a biomarker individually (Fig 7A and B), and yet together they appear to define a subtype with significantly worse prognosis (Fig 7C and D).

A second subtype with poor prognosis in LIHC was defined by *ALB-M + TP53-M*. Across all LIHC patients, those harboring *TP53-M* had shorter PFI than those with *TP53* wild-type tumors (Fig 8A). Patients harboring *TP53-M + ALB-M* had shorter PFI than those harboring either event individually (Fig 8B and C), or neither event (Fig 8D).

### *CDKN2A* and *FGFR3* co-mutation status suggest a 3-tier stratification of BLCA tumors

*FGFR3-MA + KDM6A-MD* and *FGFR3-MA + CDKN2A-MD* are the 1st and 3rd highest confidence GCRs in BLCA. The association between *CDKN2A* and prognosis is complicated in bladder cancer, as both p16 protein over-expression and complete lack of expression have been shown to be biomarkers of poor prognosis in bladder cancers (Shariat *et al*, 2004; Worst *et al*, 2018). In our study, *CDKN2A* deletion and *CDKN2A* mutations were combined into a single hybrid event, resulting in a simple association between *CDKN2A-MD* status and poor PFI (Fig 9A). We found that *FGFR3-MA* confers improved PFI within *CDKN2A* wild-type tumors (Fig 9 A), but does not associate with any PFI difference in tumors harboring *CDKN2A-MD* (Fig 9B and C). These results suggest a 3-tier classification, with *CDKN2A-MD* defining a tier with poorer prognosis, *CDKN2A/FGFR3* double wild type defining a tier with intermediate prognosis, and *FGFR3-MA⁺/CDKN2A-MD⁻* defining a tier with improved prognosis (Fig 9D).

### Comparison with SELECT

We compared the CRSO TCGA results with the pairs of co-occurrences identified by Mina *et al* (2017) using SELECT. Sixteen TCGA cancer types were analyzed by both SELECT and CRSO (Table 5). For each tissue, we compared the total coverage of all statistically co-occurrent pairs identified by SELECT within a curated set of 505 pan-cancer mutations and CNVs (Mina *et al*, 2017), versus the coverages of the core rule sets identified by CRSO (Table 5). Although both methods identified a similar mean number of synergies per cancer type (10.4 ± 10.2 for SELECT versus 11.1 ± 2.2 for CRSO), the CRSO core rules covered an average of 68% of samples (SD = 12%) per cancer type compared to an average of 19% of samples (SD = 11%) covered by the SELECT synergistic pairs. The

large discrepancy in coverage is because CRSO does not require statistical co-occurrence between the events in a cancer rule. CRSO's ability to identify a likely driver combination in the majority of samples, and to identify combinations of more than 2 events, may facilitate precision oncology advances that could benefit many patients.

For each cancer type, we compared the CRSO con-GCDs (conf. ≥ 50) with the co-occurring pairs detected by SELECT using the subset of mutation events that overlap between the datasets. Copy number events were excluded because they were processed and defined differently in Mina *et al* (Materials and Methods). In total, 99 duos were identified by either CRSO or SELECT, 5 duos were identified by both algorithms, 19 duos were identified by SELECT only, and 75 duos were identified by CRSO only (Appendix Table S7). The large discrepancy in the number of pairs is partially because the common mutation set represents a larger fraction of the CRSO event pool. Additionally, a single CRSO rule can contain 3, 6, or 10 duos if it contains 3, 4, or 5 events. Of the 19 duos identified by SELECT that were not con-GCDs, 13 had coverage below the CRSO rule-inclusion threshold of 3% (6 had coverages ≤ 1%). Of the 6 duos with coverage ≥ 3% identified by SELECT only, 4 were identified by CRSO but did not achieve 50% confidence threshold, and only 2 were not identified at all by CRSO.

CRSO identified many well-known and high coverage combinations that were missed by SELECT, including *IDH1 + {ATRX/TP53/CIC/PIK3CA}* in LGG, *BRAF + {PTEN/TP53}* in SKCM, and many high coverage combinations involving *PIK3CA* in UCEC. Many of the CRSO con-GCDs missed by SELECT consisted of a common tumor suppressor such as *TP53* and *ARID1A* cooperating with a growth-promoting oncogene. Tumor suppressors that can cooperate with many genes appear to be independent from all of them and are overlooked by approaches that rely on statistical co-occurrence. Supporting this explanation, Mina et al (2017) reported that both mutual exclusivity and co-occurrence are found at higher rates between genes that are within the same pathway. Additional factors may contribute to the differences in results between CRSO and SELECT. For example, SELECT combinations were identified across a pan-cancer cohort and then were evaluated within individual cancer types *post hoc*, whereas CRSO was applied directly to individual cancer types.

### Recurrent driver combinations across tissues

Many driver alterations are recurrently identified by GISTIC2 and dNdScv in multiple different cancer types. In order to determine whether any driver combinations are also shared across multiple cancers, we looked for overlap among the GCDs for each cancer type that achieve a minimum confidence value of 10. Across the 19 cancer types, there are 624 distinct duos and 79 of them (13%) were identified in at least two cancer types. In some cases, the SCNV regions identified by GISTIC2 in different cancers can share overlapping genes. This analysis would consider these to be distinct events, suggesting that we may be missing additional recurrences involving driver genes that appear as part of non-identical SCNV regions.

Thirty duos were identified in 3 or more cancers (Table 6). Of these, 24 contain at least one of *TP53* or *CDKN2A*, highlighting the ubiquity of these tumor suppressors across human cancers. Most of

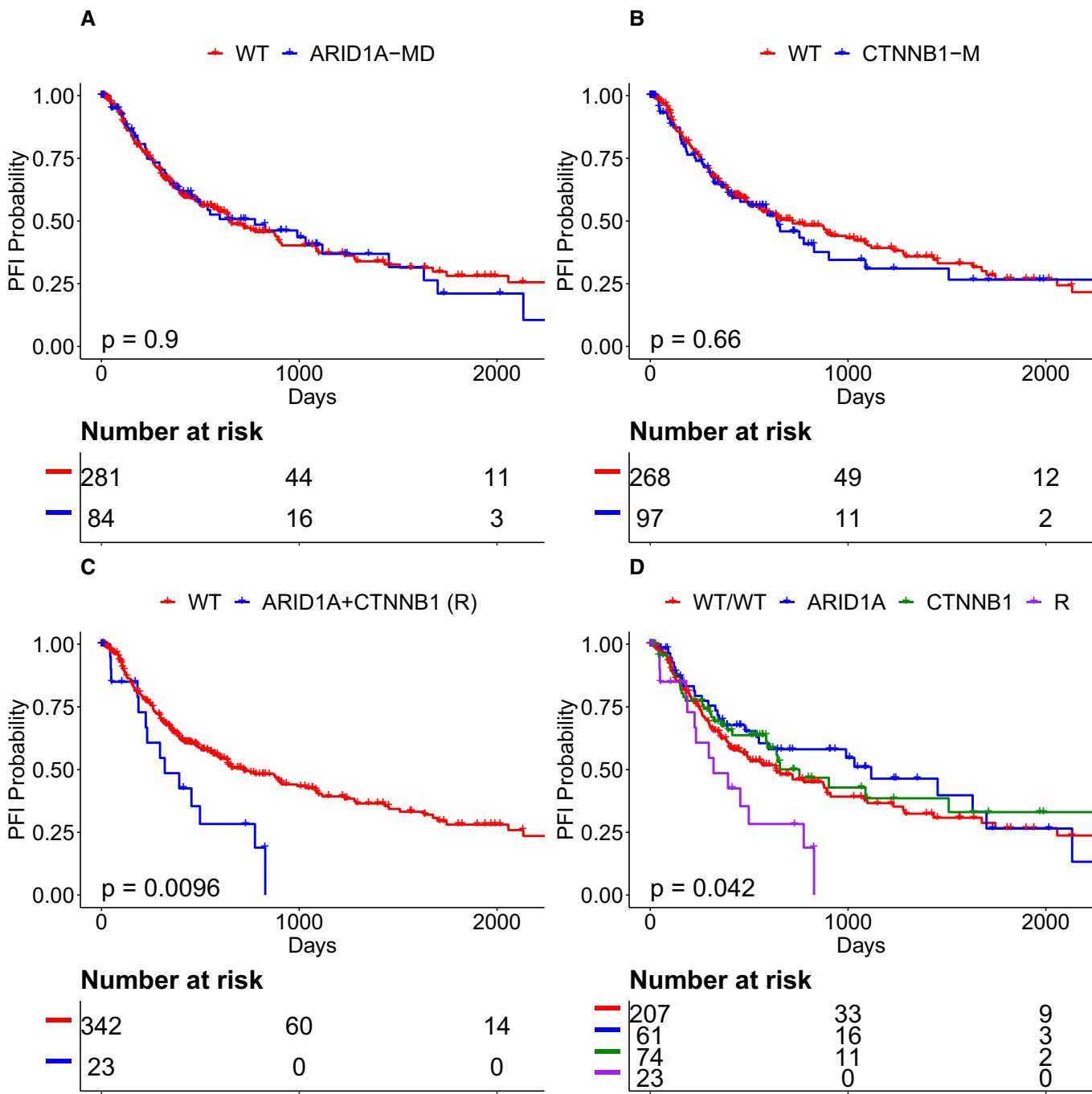

**Figure 7.  Co-occurrence of *ARID1A* + *CTNNB1* defines a subtype with poor PFI in liver hepatocellular carcinoma (LIHC).**

A    *ARID1A-MD* is not a single-gene biomarker in LIHC.

B    *CTNNB1-M* is also not a single-gene biomarker in LIHC.

C, D  LIHC patients with *ARID1A-MD* and *CTNNB1-M* define a subtype with significantly worse prognosis compared to all other patients.

Data information: *P* values calculated from Kaplan–Meier estimator.

the highly recurrent duos consist of *TP53* or *CDKN2A* paired with the well-known drivers, including *PIK3CA, PTEN, RB1, KRAS, EGFR, SMAD4, MYC,* and *BRAF.* Some of the recurrent duos in Table 6 involve lesser-known copy number events, such as *LRP1B-D, FKSG52/ MIR582/PDE4D-D, CCSER1/RN7SKP248-D, CSMD1-D, CCND1/ORAOV1-A, GMDS-D,* and *CASC8-A.* The recurrent identification of duos involving these events in 3 or more cancer types suggests that they may be more important than is currently appreciated.

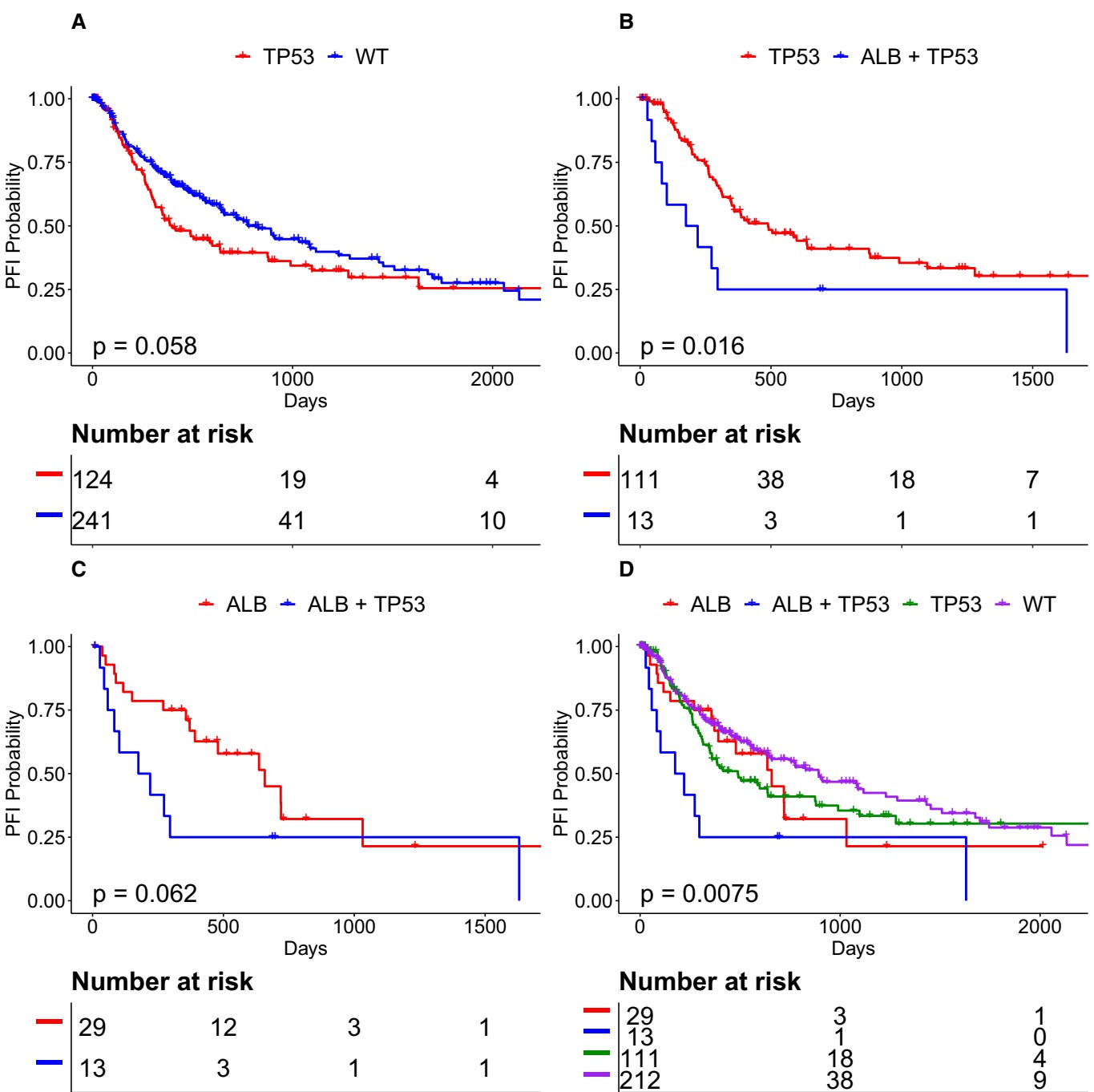

**Figure 8. Co-occurrence of *ALB* + *TP53* defines a subtype with poor PFI in LIHC.**

A       *TP53* correlates with worse PFI in LIHC.

B, C    *ALB* + *TP53* patients have worse PFI than patients with only one mutation.

D       LIHC patients with *ALB* + *TP53* define a subtype with significantly worse prognosis compared to all other patients.

Data information: *P* values calculated from Kaplan–Meier estimator.

### Comparison of TCGA results using simpler penalty matrices

The **P** matrices used in the TCGA experiments were calculated using patient-specific, event-specific, and observation-type-specific passenger probability estimations. For some user applications, it may be difficult to calculate these exact passenger probabilities. We repeated the TCGA experiments using three simplified versions of the penalty matrix to compare the results with those obtained with the original penalties. Designating the original penalty matrix as $P_O$, we define the 3 modified penalty matrices in order of increasing simplification:

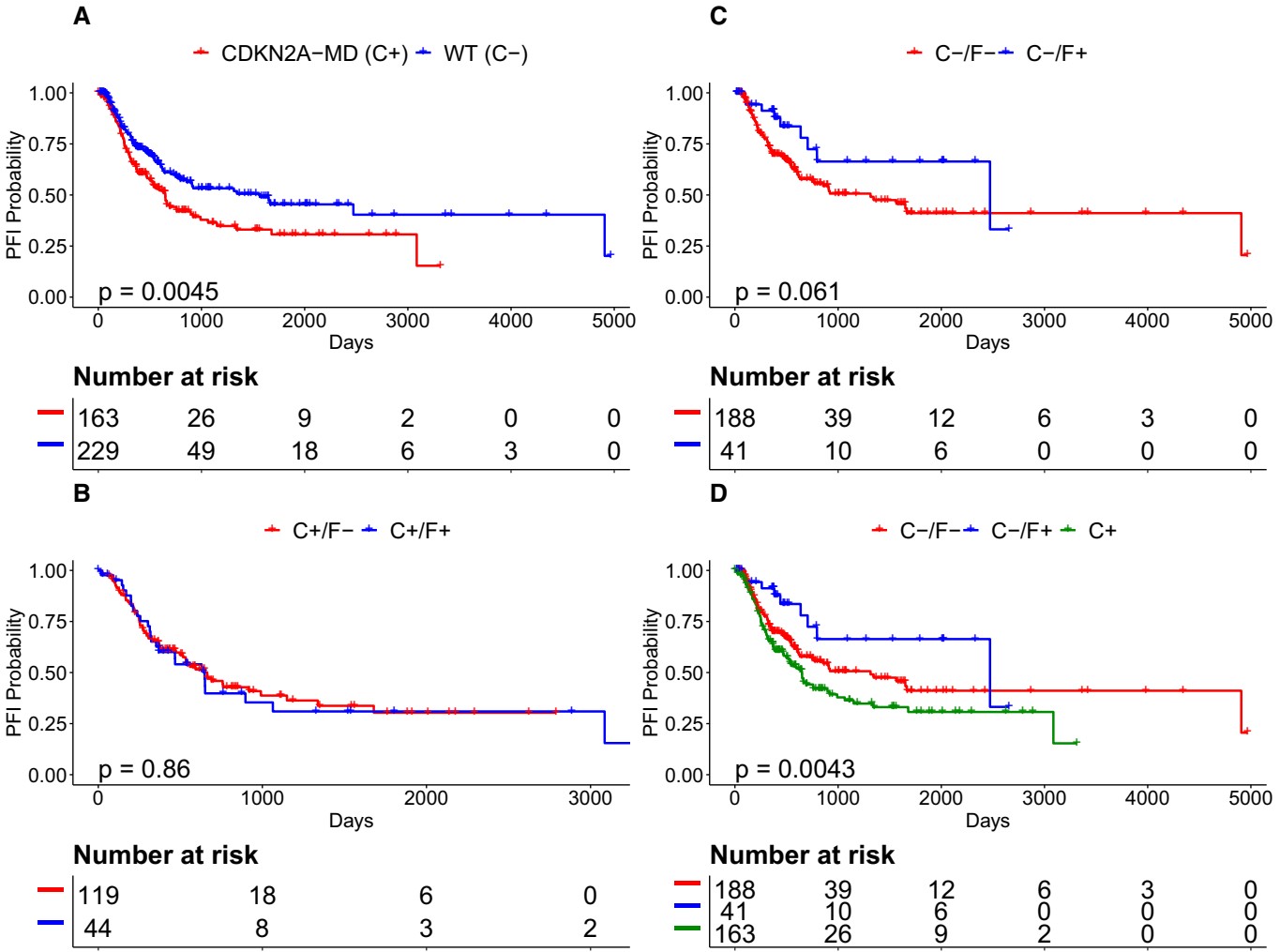

**Figure 9. BLCA outcomes depend on *CDKN2A* and *FGFR3*.**

C+/C− indicates tumors positive/wild type for *CDKN2A-MD*. F+/F- indicates tumors positive/wild type for *FGFR3-MA*.

A  *CDKN2A-MD* is a single-gene biomarker for poor PFI in bladder cancer.

B  *FGFR3-MA* is a biomarker for improved survival among CDKN2A wild-type samples.

C  *FGFR3-MA* status does not correlate with PFI in CDKN2A-MD tumors.

D  Proposed 3-tier stratification of BLCA defined by C+, C−/F−, and C−/F+.

Data information: *P* values calculated from Kaplan–Meier estimator.

- $P_{FC}$ (fixed, cancer specific) assigns a fixed penalty for each observation type (e.g., hotspot, loss, etc.) to be the mean penalty of all instances of the observation type within the given cancer type.
- $P_{FG}$ (fixed, global) uses the fixed value for each observation type across all cancers. The penalties in $P_{FG}$ were calculated by averaging over the cancer-specific mean penalties across the 19 cancer types.
- $P_U$ (uniform) uses a single fixed penalty for all events corresponding to the assumption that all non-wild-type observations have passenger probability of 0.1 (the results are the same for any fixed probability).

We ran CRSO for all 19 cancer types using $P_{FC}$, $P_{FG}$, and $P_U$ and, i.e., we compared the results with those obtained from $P_O$. For each

pairwise comparison, i.e., $P_O$ versus $P_{FC}$, $P_O$ versus $P_{FG}$, and $P_O$ versus $P_U$, we quantified the rule-agreement (Table 7) as the percentage of con-GCRs identified in either experiment that were identified in both experiments. We quantified the duo-agreement analogously (Table 7).

The mean rule-agreement was 60, 61, and 34% for $P_O$ versus $P_{FC}$, $P_O$ versus $P_{FG}$, and $P_O$ versus $P_U$, respectively, and the respective mean duo-agreements were 69, 69, and 46%. These results suggest that using a uniform fixed penalty changes the CRSO results much more than using a penalty that is fixed for each observation type. Using an observation-specific fixed penalty regardless of cancer type ($P_{FG}$) performed as well as using a cancer-specific, observation-specific fixed penalty ($P_{FC}$). The rule-agreement between $P_O$ and $P_{FG}$ was at least 75% in 8 cancer types, suggesting

**Table 5.  Coverages of SELECT pairs and CRSO core RS for 16 TCGA cancer types.**

|  | Select_Cov | CRSO_Core_Cov | Select_Num_Pairs | CRSO_Num_Rules |
|---|---|---|---|---|
| BLCA | 0.38 | 0.75 | 20 | 14 |
| BRCA | 0.27 | 0.58 | 29 | 11 |
| CESC | 0.05 | 0.53 | 2 | 12 |
| ESCA | 0.29 | 0.84 | 10 | 13 |
| GBM | 0.38 | 0.77 | 3 | 11 |
| HNSC | 0.28 | 0.68 | 9 | 10 |
| KIRC | 0.07 | 0.52 | 2 | 10 |
| LGG | 0.06 | 0.65 | 2 | 5 |
| LIHC | 0.09 | 0.56 | 0 | 12 |
| LUAD | 0.2 | 0.6 | 7 | 10 |
| LUSC | 0.15 | 0.83 | 2 | 14 |
| OV | 0.2 | 0.86 | 32 | 12 |
| PRAD | 0.1 | 0.51 | 7 | 10 |
| SKCM | 0.11 | 0.69 | 9 | 10 |
| STAD | 0.28 | 0.64 | 24 | 13 |
| UCEC | 0.11 | 0.84 | 9 | 11 |
| Average | 0.19 | 0.67 | 10.4 | 11.1 |

All SELECT predicted synergies for each cancer type were extracted, and coverage for the cancer type was calculated using the union of covered samples.

that using a fixed value for all observations of a given type can be a convenient approximation in the absence of more refined passenger probabilities. The penalties used in $\mathbf{P_{FC}}$ and $\mathbf{P_{FG}}$ are available for download.

# Discussion

We developed CRSO as a stochastic optimization procedure for predicting combinations of alterations that are minimally sufficient to drive cancer in individual patients. We applied CRSO to 19 TCGA cancer types, using SMGs identified by dNdScv (Martincorena *et al*, 2017) and SCNVs identified by GISTIC2 (Mermel *et al*, 2011) as input features. An optimal core rule set was determined for each cancer type, as well as sets of generalized core rules, trios and duos, along with confidence scores. Across 19 TCGA cancer types, CRSO prioritized biologically important events and combinations with literature support that would not be prioritized based on single-event analysis or other tools.

Several characteristics of CRSO distinguish it from other available methods for detecting biological cooperation from tumor genetic variants. Most approaches can only detect pairwise synergies, whereas CRSO is able to identify combinations of 3 or more events. CRSO accounts for passenger event probabilities by optimizing over a probabilistic representation of genetic alterations in addition to the binary representation used by most approaches. CRSO optimizes for coverage of the population and minimization of passenger penalties, using a minimal number of rules.

Although CRSO uses passenger probabilities as a basis for prioritizing events and rules, this does not mean that unassigned events should generally be considered to be passenger events that are functionally neutral. For example, in some cases essential driver events

are unassigned because they occur in samples that do not satisfy any rules in the rule library, e.g., CRSO would miss samples with only one essential driver in our dataset, suggesting that the cooperating partners for these events are missing entirely or under-represented in the dataset because of insufficient sample size (e.g., the unassigned samples in Fig 4A with *BRAF* and *NRAS* hotspots). Another scenario is that an unassigned event in a covered sample provides the tumor with selective advantages despite not having been minimally essential. More generally, CRSO is a multi-objective optimization procedure that is forced to designate only a subset of events as essential drivers because of constraints on rule set size and coverage. Without experimental validation, neither the functional consequences of individual alterations, nor the functional interactions between events can be proven by CRSO alone. Rather, we hypothesize that some of the rules identified by CRSO are of biological and clinical significance, and that the probability of a rule being of biological significance is positively correlated to its generalized core confidence score. The value of CRSO is in its ability to prioritize combinations that are likely to be biologically important from an exponentially large search space of possible combinations.

We highlighted several novel combinations from among the high confidence rules inferred by CRSO in 19 TCGA cancer types (Appendix Table S6). *NRAS-M + HULC-A* and *NRAS-M + SMYD3-A* were identified as core rules that may represent distinct mechanisms of carcinogenesis in *NRAS*-mutant melanomas. CRSO identified *NFE2L2* as con-GCRs in 4 cancer types as part of 4 distinct rules that had comparatively low coverage and single-rule performance. The CRSO findings support that *NFE2L2*-mediated NRF2 activation may be an essential driver in multiple cancers types.

Examples were presented in multiple cancer types of significant differences in patient PFIs that were found based on rules identified by CRSO. In all of these examples, the PFI differences could not

**Table 6. Generalized core duos (GCDs) identified in at least 3 cancer types.**

| Duo | Recurrence | Cancer types |
|---|---|---|
| PIK3CA + TP53 | 8 | BLCA, BRCA, COAD, ESCA, HNSC, LUSC, READ, UCEC |
| CDKN2A + TP53 | 7 | BLCA, ESCA, GBM, HNSC, LUAD, LUSC, PAAD |
| PTEN + TP53 | 7 | BRCA, COAD, GBM, LIHC, LUSC, PRAD, UCEC |
| RB1 + TP53 | 6 | BLCA, BRCA, GBM, LIHC, LUAD, LUSC |
| CSMD1-D + TP53 | 5 | BRCA, ESCA, HNSC, LUSC, READ |
| SMAD4 + TP53 | 5 | COAD, ESCA, PAAD, READ, STAD |
| KRAS + TP53 | 5 | COAD, LUAD, PAAD, READ, STAD |
| FKSG52/PDE4D-D + TP53 | 5 | ESCA, HNSC, OV, READ, STAD |
| CDKN2A + LRP1B-D | 4 | BLCA, ESCA, HNSC, LUSC |
| MYC-A + TP53 | 4 | BRCA, ESCA, OV, STAD |
| LRP1B-D + TP53 | 4 | ESCA, HNSC, LIHC, LUSC |
| CDKN2A + EGFR | 4 | GBM, HNSC, LUAD, LUSC |
| ARID1A + PIK3CA | 3 | BLCA, STAD, UCEC |
| ARID1A + TP53 | 3 | BLCA, LIHC, STAD |
| CDKN2A + NFE2L2 | 3 | BLCA, HNSC, LUSC |
| CDKN2A + PIK3CA | 3 | BLCA, GBM, HNSC |
| CCND1/ORAOV1-A + TP53 | 3 | BLCA, BRCA, LIHC |
| PIK3CA + PTEN | 3 | CESC, COAD, UCEC |
| KRAS + PIK3CA | 3 | COAD, READ, UCEC |
| KRAS + SMAD4 | 3 | COAD, PAAD, READ |
| BRAF + TP53 | 3 | COAD, LUAD, SKCM |
| FLRT3/MACROD2-D + TP53 | 3 | COAD, READ, STAD |
| GMDS-D + TP53 | 3 | ESCA, READ, STAD |
| CCSER1-D + TP53 | 3 | ESCA, READ, STAD |
| CCSER1-D + FKSG52/PDE4D-D | 3 | ESCA, READ, STAD |
| NFE2L2 + TP53 | 3 | ESCA, HNSC, LUSC |
| CDKN2A + PTEN | 3 | GBM, LUSC, SKCM |
| CASC8-A + TP53 | 3 | HNSC, LUSC, READ |
| NF1 + TP53 | 3 | LUAD, LUSC, OV |
| ARID1A + KRAS | 3 | PAAD, STAD, UCEC |

**Table 7. Comparison using simplified penalties.**

| | $P_O$ versus $P_{FC}$ | $P_O$ versus $P_{FG}$ | $P_O$ versus $P_U$ |
|---|---|---|---|
| BLCA | 67 (58) | 75 (56) | 37 (40) |
| BRCA | 79 (79) | 71 (79) | 35 (45) |
| COAD | 29 (80) | 36 (82) | 29 (72) |
| CESC | 41 (40) | 30 (31) | 23 (28) |
| ESCA | 67 (79) | 58 (72) | 5.6 (43) |
| GBM | 54 (68) | 40 (65) | 55 (62) |
| HNSC | 44 (75) | 21 (60) | 21 (40) |
| KIRC | 82 (82) | 82 (82) | 64 (64) |
| LGG | 44 (67) | 83 (89) | 29 (40) |
| LIHC | 94 (94) | 65 (65) | 24 (26) |
| LUAD | 75 (67) | 75 (62) | 43 (38) |
| LUSC | 38 (56) | 38 (57) | 29 (42) |
| OV | 77 (92) | 62 (95) | 50 (92) |
| PAAD | 100 (80) | 100 (67) | 29 (31) |
| PRAD | 50 (50) | 56 (69) | 17 (26) |
| READ | 45 (68) | 80 (91) | 12 (43) |
| SKCM | 73 (62) | 80 (73) | 70 (58) |
| STAD | 47 (62) | 59 (68) | 50 (58) |
| UCEC | 36 (47) | 56 (44) | 18 (30) |
| Mean | 60 (69) | 61 (69) | 34 (46) |

$P_O$: original penalty matrix. $P_{FC}$: simplified penalty matrix in which all observations of the same type within the same cancer are assigned a fixed value. $P_{FG}$: further simplified penalty matrix in which all observations of the same type assigned a fixed value, regardless of cancer type. $P_U$: simplest penalty matrix in which a single fixed penalty is used for all observations. Each cell shows the rule-agreement score, and the duo-agreement score is shown in parentheses.

potential biomarkers identified by CRSO may provide actionable information that goes beyond current clinical stratification, such as suggesting combination treatments or identifying specific alterations as drug targets.

CRSO is a tool for prioritizing biologically relevant combinations that may merit further investigation from a huge space of possible combinations, and it is inevitable that some identified combinations will be false positives, and some important biological combinations will be missed. The coverages achieved by the core rule sets in different cancers range from a low of 51% in prostate adenocarcinoma (PRAD) to a high of 89% in rectum adenocarcinoma (READ) cancers, revealing that a substantial subset of patients in every cancer type were not assigned to any rule. One strategy for accounting for these samples would be to expand the set of mutations and SCNVs included as events by relaxing the significance threshold used for dNdScv or GISTIC2, or by including additional events identified by other approaches. Another reason CRSO might not assign some tumors to any rule is the possibility that the tumor was driven by a single genetic alteration.

The rules identified by CRSO are further limited by the types of events that are used as inputs. We chose to use SCNVs and SMGs because these are the most common types of driver alterations in cancer, and can be systematically prioritized by statistical enrichment methods. However, many other types of alterations were

have been identified by consideration of individual alterations. Rather, these differences appear to be a consequence of specific combinations of alterations whose co-occurrences define subtypes with better or worse prognosis. These combinations, as well as other combinations in Table 4 that were not discussed, merit further investigation as prognostic biomarkers. Evaluating these findings on independent datasets is a necessary next step toward determining whether any of the combinations are reliable biomarkers that can assist clinical stratification. Incorporating information about patient treatments may help refine this analysis. The interpretability of

omitted that may contribute to cancer formation, including germline alterations, arm level CNVs, gene fusions, chromosomal translocations, and epigenetic alterations. For example, *TMPRSS2-ERG* fusions are observed in 40% of prostate cancers (Cancer Genome Atlas Network, 2015b), and p1/q19 chromosomal co-deletions are observed in 30% of lower-grade gliomas (Brat *et al,* 2015). CRSO supports inclusion of any binary-encoded event of interest. In case it is hard to calculate passenger probabilities for an event of interest, we recommend assigning a penalty equal to 1.05 times the largest penalty in each sample that harbors this event. Doing so would ensure that the event has maximum priority in the samples that harbor it. Using a much larger penalty compared to those associated with SCNVs and mutations could adversely impact the coverage of CRSO by over-prioritizing rules that cover very few samples. Although CRSO is generally robust to the exclusion of some driver events (Appendix Table S5), it is likely that exclusion of very high frequency driver events, such as *TMPRSS2-ERG* fusion, will lead to identification of some false positives.

CRSO uses the focal amplification and deletion peaks identified by GISTIC2 as input features. Some GISTIC2 peaks contain many genes, making it difficult to identify the genes of biological significance. In the absence of tools analogous to CancerEffectSizeR (Cannataro *et al,* 2018), we relied on a crude approach to estimate SCNV passenger probabilities based on alterations rates in control cytobands. Filtering out deletions and amplifications that do not have a corresponding impact on gene-expression dosage may refine the candidate driver SCNVs. Another potential refinement would be to disallow rules involving multiple co-directional copy number events that are nearby on the same chromosome, since the co-occurrence of these events may be artifacts due to proximity. Only a small fraction of all core rules were in this category, and none of them were discussed in this study.

It may be of interest to apply CRSO to established subtypes of TCGA cancer types such as melanomas that are wild type for both *BRAF* and *NRAS* or basal-like breast cancers, which have limited treatment options and poor prognosis (Bernard *et al,* 2009; Cancer Genome Atlas Network, 2012). We recommend running dNdScv and GISTIC2 on the smaller cohort to identify subtype-specific candidate drivers and running CancerEffectSizeR to quantify the effects.

Network-based stratification (NBS) is an unsupervised strategy for identifying tumor subclasses by integrating binary somatic mutation profiles with public gene–gene interaction network databases (Hofree *et al,* 2013). NBS identified subtypes that are predictive of clinical outcomes in ovarian, uterine, and lung cancers (Hofree *et al,* 2013). Liu and Zhang (2015) used NBS to identify pan-cancer subtypes characterized by different biological processes. These studies, as well as other network-based algorithms (Ciriello *et al,* 2013; Zhang *et al,* 2018), have identified tumors that are functionally similar at the pathway level and help elucidate the biological processes that characterize specific subtypes. By contrast, CRSO provides insights into specific alteration combinations that achieve pathway dysregulation in individual tumors. Future work can leverage these complementary strategies by applying CRSO within subtypes identified by network-based approaches in order to identify rules that correspond to specific functional subtypes.

Dash *et al* (2019) adapted the weighted set cover algorithm (Chvatal, 1979) to identify sets of two-hit combinations that discriminate normal and cancer samples with high accuracy. Al Hajri *et al* (2020) extended this approach to include 3-hit and some 4-hit combinations. Motivating these methods is the assumption that combinations that occur only in tumors are likely to be cooperating drivers. There are many differences in the mutational landscapes of tumors and normal tissue, and it is possible that many combinations involving passenger mutations are also likely to be observed almost exclusively in tumors. Dash *et al* considered all genes containing any coding mutation, and only identified 9 confirmed cancer genes (according to COSMIC) across the union of the 3 most frequent combinations in 17 TCGA cancer types, missing well-known drivers such as *CDKN2A, BRAF, PIK3CA, EGFR,* and many others. Specific genes were not discussed in the multi-hit version.

We developed CRSO as an approach to infer essential driver combinations that co-occur in many patients. CRSO is highly flexible and easy for researchers to use. The results represent testable hypotheses that are easy to interpret. We hope that CRSO will prove helpful in identifying biologically meaningful and clinically actionable combinations of driver alterations. From a broader perspective, we envision that wide-spread adoption of CRSO by domain experts will propel a shift in the precision oncology community from single-gene thinking toward multi-gene thinking, and that patients will be classified and treated according to driver combinations. For example, we suggest that development of a cancer combination census, analogous to the Sanger Institute Cancer Gene Census (Sondka *et al,* 2018), would accelerate therapeutic advances by nominating novel therapeutic strategies and refining clinical trial recruitment based on driver combinations. This census would be comprehensive, and include tiers based on the strength of evidence of specific combinations in specific cancer types. For example, one tier would consist experimentally validated driver combinations, such as *BRAF + PTEN* in melanomas and *ATRX + IDH1 + TP53* in gliomas, and lower tiers would consist of computationally predicted combinations nominated by methods such as CRSO and SELECT, with varying amount of literature support.

# Materials and Methods

**Reagents and Tools table**

| Reagent/Resource | Reference or source | identifier or catalog number |
|---|---|---|
| **Software** | | |
| R *version* 4.0.2 | www.r-project.org | N/A |
| RStudio *version* 1.2.5019 | www.rstudio.com | N/A |

**Reagents and Tools table**   (continued)

| Reagent/Resource | Reference or source | identifier or catalog number |
|---|---|---|
| *R packages*<br>ggplot2<br>survival<br>survminer<br>foreach<br>doMPI<br>cancereffectsizeR | Wickham (2016)<br>Therneau (2015)<br>Kassambara and Kosinski (2018)<br>Microsoft and Weston (2017)<br>Weston (2017)<br>Cannataro *et al* (2018) | N/A |
| **Data** | | |
| TCGA data Genome Data Analysis Center (2016) | (Data ref: Grossman *et al*, 2016a) | Firehose 2016 01 28 run. https://doi.org/10.7908/C11G0KM9. |
| TCGA outcome data | (Liu *et al*, 2018) | https://doi.org/10.1016/j.cell.2018.02.052 |
| **Computing resources** | | |
| Yale HPC Cluster | Yale Center for Research Computing | N/A |

## Methods and Protocols

### Representation of TCGA inputs

TCGA data from 19 cancer types (Table 1) were obtained from the January 28, 2016, GDAC Firehose (Grossman *et al*, 2016b; Data ref: Grossman *et al*, 2016a). Candidate driver mutations were defined to be the set of significantly mutated genes (SMGs) identified by dNdScv as significantly mutated using the threshold qsuball < 0.1— using tissue-specific mutational covariates developed within Cannataro *et al* (2018). Candidate copy number variations were defined to be the set of genomic regions identified by GISTIC2 as amplified or deleted, using the threshold q-residual < 0.25.

The inputs into CRSO are two event-by-sample matrices: **D** and **P**. **D** is a binary alteration matrix, such that $D_{ij} = 1$ if event *i* occurs in sample *j*, and 0 otherwise. **P** is a continuous valued penalty matrix, where $P_{ij}$ is the negative log of the probability of event *i* occurring in sample *j* by chance, i.e., as a passenger event. In order to calculate passenger probabilities, TCGA mutations and copy number variations were first represented as a categorical event-by-sample matrix, **M**. $M_{ij}$ can take one of several values called observation types. The possible observation types for event *i* differ according to the event type of event *i*. Three primary event types were considered: mutations, amplifications, and deletions. Mutations were represented at the gene level, whereas both copy number types were represented at the region level as defined by the GISTIC2 narrow peaks. Mutation events take values in the set {*Z, HS, L, S, I*}, corresponding to wild type, hotspot mutation, loss mutation, splice site mutation, or in-frame indel. Amplification events take values in {*Z, WA, SA*}, corresponding to wild type, weak amplification, and strong amplification. Similarly, deletion events take values in {*Z, WD, SD*}, corresponding to wild type, weak deletion (hemizygous), and strong deletion (homozygous).

The entries of **D** only depend on whether an observation is wild type or not. That is, $D_{ij}$ equals 0 if $M_{ij}$ is wild type, and equals 1 otherwise. By contrast, the entry $P_{ij}$ is highly sensitive to specific observation type of event *i* that is observed in sample *j*. For example, suppose *TP53* is identified as an SMG within a cancer population. Some tumors may contain one of many nonsense point mutations within *TP53*, whereas other tumors may contain a highly recurrent missense mutation, or a splice site mutation that produces an alternative isoform of the TP53 protein. Although all of these alterations are *TP53* mutations, they occur with very different passenger probabilities, sometimes spanning multiple orders of magnitude. The dual representation defined by **D** and **P** reflects the assumption that different types of alterations within the same event are functionally similar but probabilistically distinct. This representation also allows CRSO to account for sample-specific differences in passenger probabilities, which can sometimes be very substantial. Wild-type events are defined to have passenger probability of 1, so that if $M_{ij} = Z$ then $P_{ij} = 0$.

### TCGA mutational observation types

The MAF files for each TCGA dataset are annotated with many different mutation types (Appendix Table S1). To account for the fact that different kinds of mutations occur at different baseline probabilities, mutations were subdivided into four observation types: hotspots (HS), loss mutations (L), splicing mutations (S), and in-frame insertions and deletions (I).

- **Hotspot mutations:** A hotspot mutation was defined to be any SNP that leads to an alteration at a specific amino acid position that is observed in at least three samples within the population. Silent mutations and intronic mutations do not lead to amino acid changes and by definition cannot be hotspots. Most hotspot mutations are missense mutations, but the definition allows for other recurrent SNPs, such as splice site mutation or nonsense mutations to be hotspots as well. Note that the definition of hotspot does not require three instances of the exact same substitution, but rather three instances of substitutions at the same amino acid position. This choice is motivated by the fact that multiple amino acid changes in known hotspots such BRAFV600 and NRASQ61 are observed.
- **Loss mutations:** A loss mutation was defined as one occurring in a given gene if any mutation is detected except for those mutations that are silent, intronic, splice site, hotspot, in-frame insertions, or in-frame deletions. The definition of loss mutations includes missense mutations, nonsense mutations, frame-shift indels, and the other rarely observed mutations types shown in Appendix Table S1. All of these mutation types were combined under the general category of loss mutations because the majority of non-recurrent mutations will lead to loss of function.

- **In-frame indels:** In-frame indels are mutations that are in-frame deletions or in-frame insertions. In-frame insertions/deletions were categorized separately from frame-shift indels because in-frame indels have been shown to be much more likely than frame-shift indels to produce gain of function alterations (Yang *et al*, 2010).
- **Splicing mutations**: Splicing mutations are the fourth most common class of point mutation behind missense, silent and nonsense mutations (Appendix Table S1). Splicing mutations can present as point mutations within exons that lead to exon-exclusion as well as point mutations within introns that lead to intron inclusion. Because of this, many splice site mutations are not annotated with a specific amino acid change. Splicing mutations encompass only those splice sites that are non-recurrent single amino acid substitutions. When a splicing mutation occurs as a SNP at a recurrent amino acid position, it was designated as a hotspot mutation because this designation permits more accurate calculation of the associated passenger probabilities.

### Passenger probability calculation for mutations

Passenger probabilities were calculated for every observed mutation. These passenger probabilities are patient-specific, gene-specific, and observation-type-specific. Mutation rates for every possible amino acid substitution in each SMG were calculated as described in Cannataro *et al* (2018). These rates are calculated by incorporating both gene-level estimates of mutation rate (Martincorena *et al*, 2017) and tumor-type-specific mutational processes that affect nucleotide substitution rates (Rosenthal *et al*, 2016).

Hotspot and loss mutation passenger probabilities were calculated based on the mutation rates of individual amino acid substitutions calculated directly from the *cancereffectsizeR* R package (Cannataro *et al*, 2018). Hotspot mutation probabilities were calculated for each gene as the sum of the rates of all possible amino acid substitutions at hotspot positions. The loss mutation probability for a gene was calculated as the sum of the rates of all possible amino acid substitutions, except for those that occur at hotspot positions. Non-recurrent splice site substitutions were not excluded because the analysis did not include annotation of all possible splice site amino acid positions. The impact of excluding these sites will be minor, since the number of amino acids per protein is much larger than the number of splice junctions.

Control genes were used to calculate the in-frame indel and splice site probabilities. Control genes were defined to be all genes that are expressed above RSEM = 0 (RNAseq by Expectation Maximization) in at least 5% of samples and were not identified by dNdScv as SMGs. Population-level in-frame indel mutation probabilities were calculated to be the frequency of in-frame indel mutations per control gene per sample. Population-level splice site mutation rates were calculated to be the average number of splice site mutations per sample per control gene. The frequencies of in-frame indels and splice site mutations were assumed to be proportional to the number of amino acids in the protein product of each gene. A gene length adjustment factor was defined for each gene to be the number of amino acids in the gene protein product divided by 480 (approximate mean number of amino acids per protein). Gene-specific probabilities for both in-frame indels and splice site mutations were calculated to be the respective population-level probabilities multiplied by the gene length adjustment factors.

Mutation frequencies can vary greatly across patients within the same cancer type. To account for this, a patient adjustment factor was used that is based on the number of point mutations in each patient. Mutation counts for each patient were determined to be the total number of point mutations observed outside of the SMGs identified by dNdScv. Tumors for which 0 mutations were identified were assigned a mutation count of 1. In general, we do not want to remove outliers with large mutation counts because we want the penalties for observations in these samples to be down weighted accordingly. However, there were a few cases where one or two patients had such extreme outliers that they had mutation counts more than 100 times larger than the 90th percentile for the cohort. To mitigate the impact of these extreme outliers, a maximum mutation count was chosen to be the 10 times the $75^{th}$ percentile of all mutation counts. Patients with mutation counts above the maximum were assigned the maximum mutation counts. Patient adjustment factors were defined to be the patient's mutation count divided by the mean mutation count across the population. The patient-specific probabilities for every mutational observation are the product of the population-level probabilities and the patient adjustment factors.

### TCGA copy number observation types

The outputs of GISTIC2 are a set of significantly amplified copy number regions and a set of significantly deleted copy number regions. Each of the significant amplifications/deletions was represented as a single event. To do so, the copy number results were first represented at the gene level by a discrete gene-by-sample matrix of focal copy number status, $M_G$, and then the scores of individual genes within each region were combined to obtain event level features. The entries of $M_G$ take values in {$SD$, $WD$, $Z$, $WA$, $SA$}, corresponding, respectively, to strong deletions (SD), weak deletions (WD), wild type (Z), weak amplifications (WA), and strong amplifications (SA). $M_G$ was constructed by thresholding the continuous value matrix from "focal_data_by_genes.txt". Focal copy number values in [−0.3, 0.3] were designated as copy neutral, as per the noise threshold recommendation in the GDC CNV pipeline (Grossman *et al*, 2016b; Data ref: Grossman *et al*, 2016a). Values above 0.3 were designated as amplifications, and values below −0.3 were designated as deletions. To designate copy number alterations as strong or weak, the sample-specific thresholds provided in the file "sample_cutoffs.txt" were used.

The peak genes for each amplification/deletion were extracted from the tables in the files "table_amp.conf_99" /"table_del.conf_99". For each amplification peak, each sample was assigned to the maximum copy number value attained by any of the peak genes within that sample (i.e., the extreme method). Because the amplification peaks were selected for having evidence of significant amplification, amplification events are only allowed to take values in {$Z$, $WA$, $SA$}. If a deletion is observed within an amplification event, it is assigned to be wild type. This procedure results in a discrete matrix of amplification peaks by samples, $M_{AMP}$, that takes values in {$Z$, $WA$, $SA$}. Each row in $M_{AMP}$ corresponds to an amplification event identified by GISTIC2. A deletion event matrix, $M_{DEL}$, was prepared analogously. Deletion peaks were assigned to the minimum copy number value attained by any of the genes in the peak. Amplifications observed within deletion peaks were assigned to be wild type, so that $M_{DEL}$ takes values in {$SD$, $WD$, $Z$}.

### Passenger probability calculation for copy number events

It is difficult to estimate copy number probabilities at the gene level because of the strong dependence between genes that are near each other. Copy number passenger probabilities were instead estimated at the cytoband level, using control cytobands as the basis for estimating probabilities. Control cytobands were defined to be all cytobands that do not contain any genes that are within any of the significant wide regions reported in "table_amp.conf_99" and "table_del.conf_99".

A control cytoband matrix, $M_C$, was constructed by assigning each cytoband to the mode of the cytoband's genes observed in each sample from $M_G$. Consider $M_C$ to be an $n \times m$ matrix, and consider $C_{SD}$, $C_{WD}$, $C_{WA,}$ and $C_{SA}$ to be the counts of each observation type observed in the population. A population rate for each observation type was defined as follows:

$$\mu_{SD} = C_{SD}/(n*m)$$

$$\mu_{SA} = C_{SA}/(n*m)$$

$$\mu_{WD} = (C_{SD} + C_{WD})/(n*m)$$

$$\mu_{WA} = (C_{SA} + C_{WA})/(n*m)$$

The probabilities for WA and W were defined to be the frequency of observing any amplification or deletion to ensure that $\mu_{WD}$ and $\mu_{WA}$ are always larger than $\mu_{SD}$ and $\mu_{SA}$, respectively.

To account for variation in copy number rates between patients, patient-specific adjustment factors were introduced for amplifications and deletions. Let $C_j^{AMP}$ and $C_j^{DEL}$ denote the number of control cytobands amplified and deleted in patient $j$, respectively. The amplification and deletion adjustment factors for each patient were defined to be:

$$f_j^{AMP} = \left(C_j^{AMP} + 0.5\right)/\left(mean\left(\vec{C}^{AMP}\right) + 0.5\right)$$

$$f_j^{DEL} = \left(C_j^{DEL} + 0.5\right)/\left(mean\left(\vec{C}^{DEL}\right) + 0.5\right)$$

The addition of 0.5 to the counts ensures non-zero probabilities for all patients. The sample-specific probabilities for SD and WD were, respectively, calculated as $\mu_{SD,j} = \mu_{SD} * f_j^{DEL}$ and $\mu_{WD,j} = \mu_{WD} * f_j^{DEL}$. The sample-specific probabilities for SA and WA were calculated analogously.

### TCGA hybrid events

When a particular gene within a tissue type is identified by dNdScv as an SMG and is also identified by GISTIC2 as part of a SCNV, the SMG and SCNV were combined into a special event type called hybrid events. This reflects the assumption that the SCNV and SMG are exerting similar functional changes in the tumor cells. Supporting this assumption is the observation that oncogenes are frequently amplified and tumor suppressors are frequently deleted in the same cancer types. If the SCNV was a deletion, the hybrid event was denoted as "gene-MD", for mutations/deletion, and if it was an amplification, it was denoted as "gene-MA", for mutation/amplification. The hybrid events could take values in any of the mutation

types or any of the copy number types. A single hybrid event could also take two values if it was observed as an SCNV and an SMG in the same patient. For example, if *CDKN2A* loss mutation and *CDKN2A* weak deletion co-occur in a patient, this observation would have been denoted as "L,WD". In such cases, the penalty associated with the combined observation was the sum of the penalties of each observation independently (recall that penalties are -log probabilities). By increasing the penalty associated with co-occurring alterations of the same gene, the algorithm is encouraged to assign the event as a driver.

Two exceptions were encountered among the 19 TCGA cancer types that required special handling. In rectum adenocarcinoma (READ), *KRAS* was identified as part of an amplification peak and as part of a large deletion peak. Since *KRAS* is a known oncogene that is often part of amplification peaks in other cancer types, we chose to represent *KRAS* as *KRAS-MA*. The deletion peak was retained in the dataset as an ordinary deletion event, but it was not annotated as a *KRAS* hybrid event. In kidney renal clear cell carcinoma (KIRC), both *ARID1* and *MTOR* were found to be part of the same deletion peak. Since *ARID1-MD* is observed in multiple cancer types whereas *MTOR-MD* is never observed in other cancer types, we chose to represent this deletion event as part of *ARID1-MD*.

### CRSO optimization criteria

CRSO is an optimization procedure over the space of possible rule sets. The ability of a rule set to account for the distribution of events in the population is quantified by an objective function. Consider a rule set $RS = (r_1, \ldots, r_k)$, and suppose each sample has been assigned to one rule in *RS*, or to the null rule. The null rule is implicitly included in every rule set as an assignment placeholder for samples that do not satisfy any of the rules in *RS*. When a sample is assigned to a rule, the events that comprise the rule are considered to be essential drivers within that sample and therefore do not contribute to the statistical penalty under *RS*. The statistical penalty under *RS* is the sum of the penalties of all of the not assigned events under *RS*. A penalty matrix, $P^{RS}$, is derived by modifying the full penalty matrix, P, such that the penalties of all assigned events are changed to 0. For example, suppose sample $j$ is assigned to a rule containing events $x$ and $y$. This is represented in $P^{RS}$ by assigning $P^{RS}_{xj}$ and $P^{RS}_{yj}$ to be 0, instead of the original values they took in **P**.

The objective function score for *RS*, *J(RS)*, is defined to be the reduction in total statistical penalty under *RS* compared to the null rule set: $J(RS) = \sum P_{ij} - \sum P_{ij}^{RS}$ In cases when a sample satisfies multiple rules in *RS*, the sample is always assigned to the rule that maximizes *J(RS)*, i.e., the rule with largest cumulative penalty in the sample.

### Four-phase procedure for identification of best RS of size K

Given **D**, a starting rule library is built by identifying all rules that contain at least 2 events and occur in a minimum percentage of samples. Except where otherwise indicated, a minimum rule coverage threshold was chosen to be the larger of 3% of the population, or the minimum threshold that at most 2,000 rules satisfy. CRSO uses a four-phase procedure (Fig 1C) to find the best scoring rule set of fixed size, K, for $K \in \{1 \ldots 40\}$. The core rule set is subsequently chosen from among the best rule sets of size *K*. Finding the best rule set of size *K* from among $n$ rules is computationally intractable, and so we developed a heuristic procedure involving random sampling

to approximate the global optima for each $K$. The four-phase procedure involves several parameter choices that require specification. The methodology is presented using the default parameter values (Appendix Table S2) that were used for the presented applications to 19 TCGA cancer types.

- **Phase 1: Stochastic rule prioritization**. In phase 1, an iterative stochastic procedure is used to rank all of the rules in the rule library according to how likely they are to be included in the best performing rule set. For each rule, a *rule importance score* is calculated based on the average contribution of the rule within many random subsets of rules. Consider a set of rules $RS$. The contribution of rule $r_j$ within $RS$ is defined to be the percentage decrease in performance when $r_j$ is excluded from $RS$. Randomly sampled rule sets are allowed to contain family members, i.e., are not required to be valid, because we want to allow for direct competition between rules that are family members. To determine the rule importance score, multiple iterations (parameterized by *p1.spr*) of sampling multiple sizes of rule sets are evaluated separately in order to make fair comparisons between rules across a broad range of rule set sizes. For each rule set sample size, a Z score is determined from the distribution of average contributions, and the rule importance score is determined to be the average of the Z scores from different sampling sizes. The sizes of the random rule sets depend on the rule library size, as shown in Appendix Table S2. Once the importance scores are obtained for each rule in the rule pool, the 25% of rules (denoted as *p1.cut.size*) that have the smallest contribution scores are eliminated. The procedure proceeds until there are at most 24 rules, at which point the remaining rules are ranked according to a final round of importance score calculation. The purpose of using an iterative procedure rather than ranking all the rules once is to enforce direct competition among the strongest scoring rules. Our results on both simulations and real data show that the algorithm is highly robust to the choice of *p1.cut.size* and *p1.spr* and that these choices matter more for very large rule libraries.
- **Phase 2: Exhaustive rule set evaluation**. In phase 2, a subset of the top rules from phase 1 are exhaustively evaluated to determine the best rule set of each $K$. In contrast to phase 1, only valid rule sets that do not contain family members are considered in phase 2. For each $K$, the candidate rule pool is determined to be the maximum number of top rules that can be exhaustively evaluated with at most 200,000 rule sets. This computational parameter (denoted as *p2.mnrs*) was chosen to balance run time with depth of coverage. The number of rules that can be exhaustively evaluated for a given $K$ and *p2.mnrs* is referred to as the *pool size* and depends on the rule-family network between rules. For example, it is possible that no valid rule sets exist of size $K = 8$ within the top 20 rules, whereas $\approx 126,000$ valid rule sets would exist if none of the rules were family members. Since less computation is required to determine if a rule set is valid than to evaluate it, a larger computational parameter (*p2.max.compute*) is used to determine the pool sizes that lead to approximately *p2.mnrs*.
- **Phase 3: Neighbor rule set expansion.** Phase 2 results in identification of best rule sets of sizes $K = 1 \ldots 10$. However, because of computational constraints only a subset of top rules can be considered by phase 2 for inclusion among the best size-$K$ rule sets. In phase 3, the number of rules that can be included in the

top rule sets is increased by making the assumption that the global best size-$K$ rule sets will overlap highly with the size-$K$ rule sets determined from phase 2. Consider as an example that in phase 2 the best rule set of size $K = 8$ is identified from among the top $n = 30$ rules. Denote this rule set as $RS_K$. The $d_L$ neighbors of $RS_K$ are defined to be the set of all rule sets that contain $K-L$ common rules with $RS_K$. In phase 3, the search for the best performing rule sets is expanded to include rule sets that are $d_L$ neighbors of $RS_K$ and contain $L$ rules from outside of the initial top 30 rule pool, for $L = 1, 2, 3$. For each $L$, the number of new rules that can be considered is determined subject to a computational constraint (*p3.mrns*). A similar expansion of candidate rule sets is performed by considering rule sets that overlap highly with $RS_{K-1}$, allowing for consideration of new rule sets of size $K$ that contain rules that are within $RS_{K-1}$ but absent from $RS_K$. This choice is motivated by the observation the best rule set of size $K$ tends to overlap highly with the best rule set of size $K-1$.
- **Phase 4: Expansion to larger $K$**. Phase 3 results in best rule sets of sizes $K = 1 \ldots 10$ using an expanded rule pool compared to phase 2. In phase 4, the best rule set of size $K+1$ is sought using a similar procedure to the neighbor expansion of phase 3. The top 3 rule sets of size $K$ are used as seed rule sets. The candidate rule sets of size $K+1$ that are considered in phase 4 are the rule sets that share $K$, $K-1$ or $K-2$ rules in common one of the best 3 rule sets of size $K$. The maximum number of rule sets evaluated is constrained by the parameter *p4.mrns*. If for some $K'$ there are no valid rule sets that satisfy the *msa* requirement, then the algorithm stops searching for larger rule sets, and the maximum $K$ is determined to be $K_{max} = K'-1$.

### Determination of core RS and generalized core rules

In general, larger rule sets perform better than smaller rule sets because the samples in **D** have more assignment opportunities. On the other hand, larger rule sets may be more likely to reflect noise in the dataset rather than true biological signal, i.e., over-fitting. Fixing the size of the rule set enforces competition between rules based on the number of events covered per sample, the passenger rate of events covered and the number of samples covered. To mitigate the impact of over-fitting, CRSO requires that every rule in a rule set is assigned to a minimum number of samples, denoted as the *msa* parameter, for *minimum samples assigned*. The default msa parameter is 3% of the population. Rule sets that do not satisfy the *msa* threshold are automatically assigned an objective function score and coverage of 0, i.e., they are discarded.

The four-phase procedure results in a list of best rule sets of size $K$, for $K \in \{1 \ldots K_{max}\}$, where $K_{max}$ is the largest $K$ for which a valid rule set exists and satisfies the *msa* threshold. Typically, $K_{max}$ is much lower than 40, as large rule sets are constrained by the *msa* requirement as well as the requirement that rule sets do not contain family members, i.e., are valid. The goal of CRSO is to find the rule set that achieves the best balance of objective function score, sample coverage and rule set size, which is called the core rule set. The core rule set is defined to be the smallest of the best rule sets of size $K$ that achieves 90% of the maximum coverage and performance. Using a fixed threshold is a heuristic for automatically choosing a core rule set. In some cases, closer inspection of the results and the convergence curves of performance and coverage can help identify a better choice if a clear plateau is observed. To

evaluate the stability of the core rule set and automatically identify a more robust set of rules, 100 iterations of a subsampling procedure are used to identify generalized core rules (GCRs).

In each iteration, a subset of samples is randomly chosen without replacement, and a core rule set is determined by evaluating the top 100 phase 4 rule sets for each $K$. Instead of using a fixed subset size, the subset size of each iteration is uniformly drawn between 67 and 85% of the full dataset, and the core coverage and performance thresholds are both uniformly drawn to be between 85 and 99% of the maximum coverage and performance attained by any rule set over the subset. The purpose of randomizing the subset sizes and coverage/performance thresholds in each iteration is to encourage identification of rules that are robust to different experimental conditions. The full set of GCRs consists of all rules that appear in any of the subsampled cores. After performing 100 iterations, a confidence score is determined for each GCR to be the frequency of inclusion in the subcore iterations. The subset of GCRs that have confidence levels above 50 by definition cannot contain family members, and are a valid rule set. We refer to this rule set as the consensus GCR (con-GCR). The consensus GCR represents a robust estimate of the highest confidence rules for a given dataset that are most likely to reflect true biological cooperation.

In some cases, there can be a lack of high confidence GCRs, typically occurring when the core rule set contains many rules with many events. In such a situation, there may be sets of 2 events (duos) or 3 events (trios) that recur often within the subcores of each iteration. We calculate generalized core duos (GCDs) and generalized core trios (GCTs), to better identify recurrent pairwise and three-way synergies. Generalized core events (GCEs) are also calculated and represent the frequency of individual events being part of subsampled core rule sets.

### CRSO reports from 19 TCGA cancers

The CRSO reports in the Datasets EV1–EV19 provide a detailed presentation of the CRSO findings for all 19 TCGA cancer types. Each report contains seven sections, which we prefix with SR to indicate that we are referring to sections in the supplemental reports. SR Section 1 presents a basic overview of the dataset and a summary table of the parameters. Default parameter values (Appendix Table S2) were used for all cancers, with 2 exceptions: In kidney renal clear cell carcinoma (KIRC), the rule coverage threshold and *msa* were reduced to 1.4% because of sparsity of eligible rules, and in colon adenocarcinoma (COAD) the *max.nrs.p2* and *max.considered.p2* were increased to $10^6$ and $10^7$, respectively, because many of the top phase 1 rules were family members leading to a sparsity of valid phase 2 rule sets. SR Section 2 shows heatmaps of the binary matrix **D** and the penalty matrix **P** for the 20 most frequent events. SR Section 3 presents a summary of the $K$ best rule sets identified by the four-phase procedure. SR Section 4 presents a deeper dive into the core rule set. SR Section 4.1 is a table showing different characteristics of the core rules, including the coverage, phase 1 importance rank, and single-rule performance. SR Section 4.2 shows an event-by-rule breakdown of the core rule set. SR Section 4.3 shows heatmaps of **P** before and after assignment to the core rule sets. This is a visual representation of how well the rule set accounts for observations in **D**. SR Section 5 presents the generalized core analysis, consisting of GCRs, GCTs, GCDs, and GCEs, and SR Section 6 is a table summarizing the core RS rules and

the consensus GCRs, similar to Table 2. SR Section 7 is a dictionary of copy number events.

The objective function scores and coverages converged at different rates for different cancer types (Appendix Fig S8). To evaluate the impact of phases 3 and 4, we compared for each tissue the performance of the core RS post-phase 2 with the performance of the final core RS and with the performance of the core RS post-phase 3. The mean increases in J and coverage from phase 2 to phase 4 were 14 and 8.3%, respectively (Appendix Fig S9A). The mean increases in J and coverage from phase 2 to phase 3 were 1.2 and 0.6%, respectively (Appendix Fig S9B). Note that the core RS were identical post-phase 3 and post-phase 2 in 10/19 cancer types.

### Simulation methods

In order to evaluate whether CRSO could identify the ground truth rule sets, ten simulation datasets consisting of *ntr* true rules were produced for each *ntr* $\in \{2\dots 20\}$, for a total of 190 simulations. To challenge CRSO to solve problems similar to those presented by the real data, the passenger probability distributions, rule size distribution, and ground truth rule set network structure were informed by the characteristics of the 19 TCGA input datasets, and the CRSO results for these cancer types.

The size of each synthetic rule was sampled from a distribution of rule sizes, $\vec{d_s}$ of all 194 aggregated consensus-GCRs, so that each synthetic rule consisted of 2, 3, 4, 5, or 6 events with approximate probabilities 73, 19, 4, 3, and 1% respectively. The distributions of events across the con-GCRs in the TCGA datasets were non-uniform, with some events being part of many rules and many events being part of a few rules. We therefore defined an event inclusion probability vector, $\vec{p}_{\text{lnc}}$, by averaging over the distribution of the ordered event inclusion fractions over the set of cancer types that comprise between *ntr*−5 and *ntr*+5 consensus GCRs, and then normalizing to sum to 1. The following procedure was used to construct a simulated dataset of size *ntr*:

1 Determine the ground truth rule set $RS_{GT}$:

  a Make rule 1:
   i Choose number of events in rule, *rs*, by sampling from $\vec{d_s}$
   ii Sample *rs* distinct events from the pool of events using probabilities $\vec{p}_{\text{lnc}}$
  b For rule *j* in 2 through *ntr*:
   i Make a random rule as in step 1
   ii If rule *j* is distinct and not a family member with any prior rules proceed. Otherwise restart rule *j*.

2 Assign samples to rules in $RS_{GT}$ and populate a matrix of true drivers, denoted by $\mathbf{D_0}$:

  a Determine the rule coverage distribution for the rules in $RS_{GT}$ such that each rule is assigned to at least 3% of samples. First each rule is assigned probability 3% and then the remaining probability is added by randomly partitioning the excess probability interval.
  b Determine a fraction of rules uniformly sampled between 0.01 and 0.20 that will be assigned to the null rule.
  c Initialize $100 \times 400$ matrix $\mathbf{D_0}$ representing the ground truth drivers in each sample. Assign samples to rules or to the null

rule according to the above probabilities. $\mathbf{D_0}$ *(i, j) = 1* if and only if sample *j* is assigned to a rule containing event *i*. Otherwise $\mathbf{D_0}$ *(i, j) = 0*.

3  Add noise sampled from empirical distributions to populate simulation matrix **D** and **P**:

a  Determine event-specific passenger rates, $\vec{p_e}$, by sampling with replacement from the pool of TCGA passenger event rates. The pool of TCGA passenger event rates consisted of the pooled frequencies of all non-driver events in each cancer type, where driver events are any event that is part of at least one con-GCR. Passenger event rates below 0.02 were excluded from the pool in order to make the simulation more challenging to CRSO.

b  Determine sample adjustment factors, $\vec{a_s}$, from the pool of TCGA passenger sample adjustment factors. TCGA passenger sample adjustment factors for each cancer type were defined as the fraction of non-driver events in each sample normalized across the tumor population to have mean 1.

c  Define a passenger probability matrix, $\mathbf{P_{prob}}$, so that $P_{prob}(i, j) = \min(p_e(i) * a_s(j), 0.95)$.

d  Use $\mathbf{P_{prob}}$ to populate **D** and **P**. For entry *(i, j)*:
   i   If $\mathbf{D_0}$ *(i,j) = 1*, then $D(i,j) = 1$ and $P(i,j) = -\log(P_{prob}(i,j))$.
   ii  If $\mathbf{D_0}$ *(i, j) = 0*, then generate a uniform random value *u*. If $u \leq P_{prob}(i, j)$, then $D(i, j) = 1$ and $P(i,j) = -\log(P_{prob}(i,j))$. If $u > P_{prob}(i,j)$, then $D(i,j) = 0$ and $P(i,j) = 1$.

CRSO was applied to each simulation with the default parameters (Appendix Table S2), except for a few modifications to reduce computational cost. The *p1.ntpr* was reduced from 40 to 20, max rule sets evaluated for phases 2, 3, and 4 were reduced by 50%, and the number of GC iterations was reduced from 100 to 40. The upper limit for the max library size was removed. The starting rule library sizes for the 190 simulations ranged from a low of 192 rules to a high of 5,447 rules, and were between 350 and 2,000 for 182 out of 190 simulations (Appendix Fig S4).

### Phase 1 performance analysis and parameter optimization

We evaluated the performance of phase 1 on the simulations in order to assess whether alternative parameter choices would improve CRSO's overall performance. Phase 1 prioritizes rules based on their contribution to random rule sets, and is critical for reducing the search space of possible rule sets, which would otherwise be computationally intractable. This is an inherently heuristic strategy whose performance is mathematically difficult to evaluate without knowledge of the ground truth rule set. A phase 1 score was defined for each of the 190 simulations to quantify the similarity of the phase 1 ranking to the maximum possible ranking, corresponding to all true rules in the top *ntr* positions:

Given a rule library of *RL* rules, the phase 1 results can be represented as a ranking of rules, denoted by $\vec{r_{p1}}$. As a baseline for comparison, we order the rules in the rule library according to decreasing coverage across the population, which we denote by $\vec{r_{cov}}$. Given knowledge of the ground truth rule set and a ranking, $\vec{r}$, we define $y(\vec{r}, x)$ to be the fraction of ground truth rules identified by position *x*, for all $x \in \{1 \dots RL\}$. We then define $AUC(\vec{r}) = \sum y(\vec{r}, x)$ from x = 1 to $x_{max}$, where $x_{max}$ is the minimum

between 50 and the position of the last ground truth rule within $\vec{r_{cov}}$. The maximum possible *AUC* corresponds to a ranking in which all *ntr* true rules are ranked in the top *ntr* positions, and we denote this by $AUC_{max}$. The *AUC* of the frequency ranking is defined to be a baseline *AUC*: $AUC_0 = AUC(\vec{r_{cov}})$. Finally, we define the phase 1 score of the ranking $\vec{r_{p1}}$ to be $S_{p1}(\vec{r_{p1}}) = (AUC(\vec{r_{p1}}) - AUC_0)/(AUC_{max} - AUC_0)$. In case $AUC_0 = AUC_{max}$, we define $S_{p1}(\vec{r_{p1}}) = AUC(\vec{r_{p1}})/AUC_{max}$. This happened only 3 times in the 190 simulations, and *ntr* equaled 2 for all 3 cases.

The mean phase 1 score was more than 0.94 for all ground truth rule set sizes, indicating that phase-1 rankings successfully prioritized the ground truth rules (Appendix Table S4). Phase-1 rankings were far superior at prioritizing ground truth rules compared to coverage or SJ rankings for all *ntr*. In 186 out of 190 (0.98) total simulations, 100% of ground truth rules were identified within the top 40 rules post-phase 1. By comparison, this was the case for 30 simulations (16%) using coverage rankings and for 19 simulations (10%) using single-rule performance (SJ) rankings. On average, 99.8% of true rules were identified among the top 40 phase 1 rules, compare to 64% for coverage rankings and 58% for SJ ranking (see Appendix Table S4 for breakdown by *ntr*).

The simulation results suggest that the phase 1 procedure is very effective at prioritizing ground truth rules using the default parameters of *p1.ntpr* = 20 and p1.cs = 0.25. In order to understand how the parameter choices impact the phase 1 performance, we performed phase 1 over a grid of parameter values for each of the 190 simulations. Specifically, we evaluated phase 1 using a grid of parameter pairs *(p1.cutsize, p1.ntpr)*.

First we compared the performance of *p1.cutsize* = 1, corresponding to all of the rules ranked in a single pass through, versus *p1.cutsize* = 0.5, corresponding to 50% of rules being eliminated in each iteration. We found that *p1.cutsize* = 0.5 was statistically superior over the 190 simulations for each *p1.ntpr* $\in$ *{1, 2, 10, 20, 40}*. We also found that was *p1.ntpr* = 10 is significantly better than *p1.ntpr* $\leq$ 2 in for all *p1.cutsize* values. There was no statistical performance difference between *p1.ntpr* = 10 and either *p1.ntpr* = 20 or 40 for any cut size. There was no statistically significant difference between *p1.cutsize* pairs (.50, .25), (.25, .10), or (.50, .10) for any *p1.ntprs* > 1. There was no statistical difference between (*p1.cutsize* = 0.5, *p1.ntpr* = 10) and (*p1.cutsize* = 0.1, *p1.ntpr* = 40). We can conclude that a pairing of (*p1.cutsize* = 0.5, *p1.ntpr* = 10) is optimal since no benefit is gained by reducing *p1.cutsize* or increasing *p1.ntpr*, both of which incur a computational cost. Accordingly, we conclude that the parameters used in the simulations and in the TCGA experiments were sufficient to optimize phase 1 performance, but the same level of performance could have been achieved using less computation.

### Robustness of CRSO to missing features

The rules that can be identified by CRSO depend on the collection of events included in the inputs. In order to test the robustness of CRSO to the exclusion of important features, CRSO was applied to TCGA melanoma data excluding a single event, for each of the top 15 events. The results were compared to the results from the full dataset. We calculated several metrics for each event excluded. First, we summarize the excluded event *E* by calculating: (i) frequency in the population, (ii) percentage of GCDs containing *E*, and (iii) percentage of weighted GCDs, calculating by summing

over the confidence scores of the GCDs that contain *E* and dividing by the sum of all GCD confidence scores. These are reflections of the event's importance to the full results, and can be thought of as unavoidable losses in the case of missing events (see Appendix Table S5, columns 2–4).

Next we calculate the agreement of the smaller dataset (i.e., dataset with event excluded) with the full dataset results that do not contain the event. Consider the truth to be the full input results. Eligible duos are those duos identified in the full results that do not contain *E*, since it is possible for these duos to be identified in the event-excluded results. Retention is the percentage of eligible duos from the full results that are also identified in the event-excluded results. Weighted retention accounts for the confidence scores of the retained/not retained rules, and is calculated by summing over the confidence scores of all retained duos and dividing by the sum of the confidence scores of all eligible duos.

Rules or duos that are missing from the full dataset can be gained in the smaller dataset results. Considering the full input results to be the ground truth, these gained duos are false positives. The false-positive rate (FPR) for each exclusion experiment is the percentage of smaller dataset results that were identified in the full dataset results. Weighted FPR is the summed confidence scores of the gained duos divided by the summed confidence scores of all duos identified by the smaller dataset.

The melanoma experiment found that weighted retention was over 93% for all exclusion experiments (Appendix Table S5). The weighted FPR was below 10% for all events except for *CDKN2A-MD* and *NRAS-M*, both of which are high frequency, highly included events. Overall, the results suggest that if a driver event is excluded from the inputs, the duos/rules that do not contain the event will generally be comprehensively captured in the incomplete dataset. The rate of false associations caused by the event exclusion is generally small for most events, but can be large if the excluded event is frequent and part of many duos/rules in the full dataset.

### Criteria for common event comparison with SELECT

Sixteen cancer types were included in the comparison between CRSO and SELECT (Mina *et al*, 2017). COAD and READ were excluded from the side-by-side comparison because they were analyzed as a single colorectal cancer type in Mina *et al*. Pancreatic adenocarcinoma (PAAD) was excluded because it was not analyzed in Mina *et al*. For each cancer type, a set of common mutations was identified as the intersection of the SELECT pan-cancer mutations with the cancer-specific mutation events identified by dNdScv that were used in the CRSO analysis. *CDKN2A* mutations were excluded from the common set of mutations in cancers for which *CDKN2A* was identified as hybrid mutation/deletion in the CRSO analysis. In these cancers, large coverage discrepancies were observed in duos involving *CDKN2A* because SELECT encoded *CDKN2A* mutations as separate events from *CDKN2A* copy number loss. Common event duos for each cancer were defined to be the union of GCDs identified by CRSO and the co-occurrent pairs identified by SELECT for which both events were in the common set of mutations.

### Frequently used acronyms

- CRSO Cancer Rule Set Optimization.
- TCGA The Cancer Genome Atlas.

- SCNV Somatic copy number variation.
- SMG Significantly mutated gene.
- dNdScv Name of method for identifying significantly mutated genes.
- GISTIC2 Name of method for identifying significant copy number variations.
- GCR Generalized core rule.
- GCT Generalized core trio.
- GCD Generalized core duo.
- GCE Generalized core event.
- GC Generalized core (sometimes used in other contexts such as GC iterations).
- con-GCR Consensus generalized core rule, i.e., GCRs with confidence > 50%.
- RS Rule set.
- MSA Minimum samples assigned. All rules in a rule set must be assigned to at least msa samples.
- NTR Number of true rules. Used to describe ground truth rule set size of simulation.
- PFI Progression-free interval.

TCGA acronyms:

- SKCM Skin cutaneous melanoma.
- LIHC Liver hepatocellular carcinoma.
- COAD Colon adenocarcinoma.
- READ Rectum adenocarcinoma.
- LGG Low-grade glioma.
- HNSC Head and neck squamous cell carcinoma.
- BLCA Bladder urothelial carcinoma.

## Data availability

- The CRSO R code is available at https://github.com/mikekleinsgit/CRSO/
- The preprocessed TCGA datasets that were used in this study are available at https://github.com/mikekleinsgit/preprocessed-crso-tcga-datasets/. The simplified tissue-specific and pan-cancer penalties are available in the file "mean_tissue_penalties.RData"

**Expanded View** for this article is available online.

### Acknowledgements

The results published here are based on data generated by the TCGA Research Network: http://cancergenome.nih.gov/ (Grossman *et al*, 2016b). The authors thank Dr. Michael Kane for his help with developing the CRSO software package. The authors thank Dr. Christos Hatzis for his counsel. This work was supported by National Institutes of Health (grant numbers P30CA016359 to HZ, P50CA1965305 to HZ).

### Author contributions

MIK, DFS, and HZ conceived the project. MIK programmed the CRSO software and drafted the manuscript. VLC and JPT provided SNV mutation rates and dNdScv values that were used to calculate TCGA passenger probabilities. SN reviewed the TCGA results and identified interesting and potentially novel combinations via literature exploration. JPT, SN, DFS, and HZ edited the manuscript. All authors read and approved the final manuscript.

## Conflict of interest

The authors declare that they have no conflict of interest.

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
