## [Review Process File · Molecular Systems Biology]

Identifying Modules of Cooperating Cancer Drivers

Michael Klein, Vincent Cannataro, Scott Newman, Jeffrey Townsend, David F Stern, and Hongyu Zhao

DOI: [10.15252/msb.20209810](https://doi.org/10.15252/msb.20209810)

Corresponding author(s): Hongyu Zhao (hongyu.zhao@yale.edu) , David F Stern (DF.Stern@yale.edu), Michael Klein (miklein87@gmail.com)

Review Timeline:

Submission Date:	23rd Jun 20
Editorial Decision:	24th Aug 20
Revision Received:	18th Nov 20
Editorial Decision:	7th Jan 21
Revision Received:	20th Jan 21
Accepted:	26th Jan 21

Editor: Maria Polychronidou

Transaction Report:

Thank you again for submitting your work to Molecular Systems Biology. First of all, I would like to apologise for the exceptional delay in getting back to you. Unfortunately, after a series of reminders, reviewer #3 never returned their comments. In the meantime we have now heard back the other two referees and in the interest of time we have decided to proceed with making a decision based on these two reports. As you will see below, the reviewers acknowledge that the presented approach is potentially interesting. However, they raise a series of concerns, which we would ask you to address in a major revision.

I think that the recommendations of the reviewers are rather clear and there is therefore no need to repeat the points listed below. All issues raised by the referees would need to be satisfactorily addressed. Please let me know in case you would like to discuss in further detail any of the issues raised.

On a more editorial level, we would ask you to address the following issues.

REFeree REPORTS

Reviewer #1:

Klein et al report a new method to identify combinations of genomic alterations that together act as drivers of tumorigenesis. Conceptually, this is an important addition to the field of cancer driver identification, with the potential to move the field beyond the identification of single gene drivers towards a multi-gene model, likely with greater biological validity.

The term 'specificity' is misused throughout the manuscript. The correct statistical term to refer to $TP/(TP+FP)$ is 'precision'.

The term 'passenger' is also used in an unintuitive manner. If I've understood the manuscript correctly, all genomic aberrations included are known drivers. Aberrations that are referred to in the paper as 'passengers' are in fact driver events that do not form part of a cooperative module within a particular sample.

Related to the previous point, I assume that CRSO is not helpful in identifying drivers in single-gene diseases, such as many pediatric cancers. This should be mentioned in the Discussion.

In figure 1C, K2 and K4 are undefined. Also, K2 is referred to twice. Should one of these values be labelled as K3?

On p.10, the authors report all rules with p-values below 0.05 before multiple testing correction. It would be more appropriate to report the rules whose adjusted p-values pass a threshold.

The description of subsampling on p.22 is unclear. Why is there a range (67%-85%) of samples included in each iteration, rather than a fixed value? Are samples chosen with or without replacement?

Similarly, the comment, 'core coverage and performance thresholds are randomly sampled uniformly between 85-99%' is unclear. Does this mean that thresholds are sampled at regular intervals (85%, 86%, *7% ...) or that thresholds are sampled randomly from a uniform distribution between 85% and 99%?

The choice of the best rule set size, K, is described on p.22 as 'arbitrary'. It would be useful if the authors provided additional guidance concerning situations where users would want to override the standard algorithm and pick another K value.

Reviewer #2:

The work by Klein and colleagues is an intriguing exercise aimed at identifying oncogenic combinations of alterations which characterize a subset of genetically similar tumors. The authors propose a rigorous approach to identify the minimal set of combinations that explain the maximal

fraction of patients within a given cohort. In these terms, the authors promote the idea (which I completely espouse) that oncogenic alterations need to be analyzed and considered in a combinatorial manner rather than individually and independently.

The algorithmic approach could be of use to characterize the genetic features of a given tumor cohort, and indeed it would have been nice to have this approach years ago while the TCGA project was running. The results are interesting, although several are known, and the manuscript is well written, even though I would encourage the authors to substantially reduce the use of acronyms and jargon which at times make the text hard to read.

My major concerns are mostly conceptual and on the interpretation of certain results, plus a few methodological clarifications that I feel need to be provided.

1) The main conceptual concern that I have is in the way this method is proposed. The authors begin discussing the relevance of mutually exclusive and cooccurring alterations in cancer, citing the relevant literature (which I'm afraid is actually much vaster, especially for mutual exclusivity) and comparing their approach with a recent algorithm to detect significant mutual exclusivity and cooccurrence.

However, as stated by the authors themselves, their method does not search for statistical significant cooccurrence (see first paragraph pg. 16), but for a minimal set of combinations able to cover the majority of alteration occurrences (in this way concurrent alterations are favored, but the cooccurrence is not necessarily unexpected). This is not a flaw, it could actually be an advantage of this approach, but the comparison with other tools to detect statistically significant cooccurrence is not proper.

This tool is in line with other approaches designed to stratify patients based on minimal subsets of recurrent alterations, see e.g.

<https://www.ncbi.nlm.nih.gov/pmc/articles/PMC4491878/>

<https://www.nature.com/articles/ng.2762>

<https://www.nature.com/articles/nmeth.2651>

<https://pubmed.ncbi.nlm.nih.gov/29949979/>

in all these cases, as in the algorithm presented by the authors, cooccurrence is intrinsically favored by the algorithms but is not assessed per se.

Given the algorithms above have been often applied to the TCGA cohorts (either in their respective publications or directly in the marker studies from TCGA) the authors should compare their results with results obtained by those approaches and discuss their tool in the context of these and other similar methods.

2) The other major issue I have is with the interpretation of some of the results. There are two strong assumptions in the way the manuscript is currently written which are not substantiated:

- First, once a final set of rules is defined for a tumor type, alterations in each patient are classified as driver or passenger based on them being part or not of the rule to which the patient was assigned. I understand this is just nomenclature (or at least I hope it is) but the authors cannot conclude that an alteration in a given patient is non-functional simply because it is not found in the combination predicted by the algorithm.

E.g. in figure 4A, several BRAF mutations are classified as passenger: what is the evidence for that?

If the authors can effectively provide data supporting a different functional effect for these mutations, then it would be incredibly interesting to demonstrate that their algorithm, as an added bonus, can indeed discriminate functional and non-functional events. However, if they can't, the terminology need to be changed and it should be made clear in the text that a rule-patient assignment does NOT imply that only alterations in the rule are functional.

- Second, the authors did a commendable effort to survey the literature to find support of functional interactions between concurrent alterations.

I'm not sure, however, the combinations detected by this method need to reflect functional interactions.

As mentioned in my previous comment, the cooccurrence detected by this tool is not necessarily unexpected (i.e. statistically significant), it might simply reflect tumor subtypes associated with cell of origin (e.g. see the glioma rules, once histological or molecular subtypes are accounted for, the cooccurrence is trivial - which is why they were not found by SELECT), differentiation status or other molecular modifications.

It is in general difficult to establish a chain of causation among the molecular features and phenotypes observed in cancer cells,

thus implying that concurrent events in a rule defined by their algorithm are as such because of underlying functional relationships is here over-reaching. The authors should tone down certain statements and acknowledge that several covariates are here not accounted for.

3) It would have been compelling to have this tool running on a cohort combining all tumor types. Was this not done because of efficiency issues?

This would have been a great analysis to show how much tumor subtypes can be defined by combination of alterations, oblivious of their sites of origin. Similar analyses have been previously done, it is a pity not having it here.

If indeed efficiency issues prevent this analysis, then I would strongly suggest to include a comparative analysis of rules determined across tumor types, e.g. to propose tumor type specific and pan-tumor rules.

4) From a methodological perspective: what is effect of the scores computed in the P matrix to the final rule set?

By absurd, if all values in P were set to 1 when an alteration occur in a sample (0 otherwise), would the results obtained be dramatically different?

i'm asking because the computation of P is quite laborious and might represent a bottleneck to a broad application of this approach, but at the same time the objective function is based on it.

The authors should investigate what is the effect of using simpler versions of P, e.g. having different fixed scores for missense, loss-of-function and hotspot mutations. The way it is currently done is more sophisticated, but how much does it help?

It would be important to know.

5) On a similar tone, once a core rule set is identified, the approach goes through a lengthy series of refinements. what is the effect of these refinements in terms of result? Do they significantly improve the performance? (coverage? number of rules?)

Reviewer #1:

Klein et al report a new method to identify combinations of genomic alterations that together act as drivers of tumorigenesis. Conceptually, this is an important addition to the field of cancer driver identification, with the potential to move the field beyond the identification of single gene drivers towards a multi-gene model, likely with greater biological validity.

1. The term 'specificity' is misused throughout the manuscript. The correct statistical term to refer to $TP/(TP+FP)$ is 'precision'.

We thank Reviewer 1 for bringing this mistake to our attention. We have made this correction throughout the manuscript.

2. The term 'passenger' is also used in an unintuitive manner. If I've understood the manuscript correctly, all genomic aberrations included are known drivers. Aberrations that are referred to in the paper as 'passengers' are in fact driver events that do not form part of a cooperative module within a particular sample.

We thank Reviewer 1 for bringing this important point to our attention. We changed this confusing terminology, please refer to major change 1 above.

3. Related to the previous point, I assume that CRSO is not helpful in identifying drivers in single-gene diseases, such as many pediatric cancers. This should be mentioned in the Discussion.

We thank Reviewer 1 for this comment. We are uncertain about the precise meaning of “single-gene disease” in this context.

We hope that Reviewer 1’s concerns will be assuaged by the following excerpts from the discussion section, the first of which is a new addition:

1. “Although CRSO uses passenger probabilities as a basis for prioritizing events and rules, this does not mean that unassigned events should generally be considered to be passenger events that are functionally neutral. For example, in some cases essential driver events are unassigned because they occur in samples that do not satisfy any rules in the rule library, suggesting that the cooperating partners for these events are missing entirely or under-represented in the dataset because of insufficient sample size (e.g., the unassigned samples in Figure 4A with *BRAF* and *NRAS* hotspots).”
2. “However, many other types of alterations were omitted that may contribute to cancer formation, including germline alterations, arm level CNVs, gene fusions, chromosomal translocations and epigenetic alterations.”

4. In figure 1C, K2 and K4 are undefined. Also, K2 is referred to twice. Should one of these values be labelled as K3?

Thank you for pointing out this omission from the caption. We realized that introducing the variable names K2 and K4 was unnecessary and confusing. Instead, we have replaced the variables with the actual parameter values that were used in the TCGA experiments and are the default values: 10 for K2 and 40 for K4 (10 is used as the maximum in both phases 2 and 3).

5. On p. 10, the authors report all rules with p-values below 0.05 before multiple testing correction. It would be more appropriate to report the rules whose adjusted p-values pass a threshold.

Thank you for this suggestion. We have modified the text and Table 4 to report associations that satisfy $P_{Adj} \leq 0.15$ as well as $Cox-PH P \leq 0.05$. We also moved the text explaining the permutation procedure used to calculate P_{Adj} from the Methods subsection “Clinical outcome analysis” directly into the results subsection “Associations of rules with patient outcomes”. We did this for two reasons: 1) we felt it was important to explain to the reader that we used a very conservative permutation approach in order to explain the large discrepancies between $Cox-PH P$ and P_{Adj} that were reported for some of the rule vs. event comparisons. 2) We were able to explain the clinical outcome procedure entirely by adding only 2-3 extra sentences into the results, allowing us to remove the 2 paragraph “Clinical outcome analysis” subsection from the Methods altogether (since there was a lot of redundancy between the descriptions in the results and in methods).

6. The description of subsampling on p.22 is unclear. Why is there a range (67%-85%) of samples included in each iteration, rather than a fixed value? Are samples chosen with or without replacement? Similarly, the comment, 'core coverage and performance thresholds are randomly sampled uniformly between 85-99%' is unclear. Does this mean that thresholds are sampled at regular intervals (85%, 86%, *7% ...) or that thresholds are sampled randomly from a uniform distribution between 85% and 99%?

Thank you for this comment. We added text to clarify the subsampling procedure and explain our motivation for using ranges rather than fixed values.

Here is the new text, with changes highlighted in red:

“In each iteration, a subset of samples is randomly chosen without replacement, and a core rule set is determined by evaluating the top 100 phase 4 rule sets for each K. Instead of using a fixed subset size, the subset size of each iteration is uniformly drawn between 67-85% of the full dataset, and the core coverage and performance thresholds are both uniformly drawn to be between 85-99% of the maximum coverage and performance attained by any rule set over the subset. The purpose of randomizing the subset sizes and coverage/performance thresholds in each iteration is to encourage identification of rules that are robust to different experimental conditions. The full set of GCRs consists of all rules that appear in any of the subsampled cores. After performing 100 iterations, a confidence score is determined for each GCR to be the frequency of inclusion in the sub-core iterations. The subset of GCRs that have confidence levels above 50 by definition cannot contain family members, and are a valid rule set. We refer to this rule set as the consensus GCR (con-GCR). The consensus GCR represents a robust estimate of the highest confidence rules for a given dataset that are most likely to reflect true biological cooperation.”

And here is the old text for reference:

“In each iteration, between 67-85% of the samples are randomly chosen, and a core rule set is determined by evaluating the top 100 phase 4 rule sets for each K. The *msa* is scaled according to the subset size, and the core coverage and performance thresholds are randomly sampled uniformly between 85-99%. The full set of GCRs consists of all rules that appear in any of the subsampled cores. After performing 100 iterations, a confidence score is determined for each GCR to be the frequency of inclusion in the sub-core iterations. The subset of GCRs that have

confidence levels above 50 by definition cannot contain family members, and are a valid rule set. We refer to this rule set as the consensus GCR (con-GCR). The consensus GCR represents a robust estimate of the highest confidence rules for a given dataset that are most likely to reflect true biological cooperation.”

7. The choice of the best rule set size, K , is described on p.22 as 'arbitrary'. It would be useful if the authors provided additional guidance concerning situations where users would want to override the standard algorithm and pick another K value.

Thank you for this comment. We modified the text as shown below. We changed the word “arbitrary” to “fixed”. The situation in which a user would consider overriding the standard K is if the slope shows a clear flattening that occurs slightly below the threshold of 90%, e.g., $K-1$ would be a better choice if it achieves 89.9% and K achieves 90.1%. However, we decided to remove this suggestion from the revised manuscript, since: 1) it may be confusing to the reader and 2) the generalized core procedure was designed to account for cases where such plateauing occurs by randomizing the thresholds and subsampling the dataset.

Here is the new text, with changes highlighted in red:

“The core rule set is defined to be the smallest of the best rule sets of size K that achieves 90% of the maximum coverage and performance. Using a fixed threshold is a heuristic for automatically choosing a core rule set. To evaluate the stability of the core rule set and automatically identify a more robust set of rules, 100 iterations of a subsampling procedure are used to identify **generalized core rules** (GCRs).”

And here is the old text for reference:

“The core rule set is defined to be the smallest of the best rule sets of size K that achieves 90% of the maximum coverage and performance. Using an arbitrary threshold is a heuristic for automatically choosing a core rule set. In some cases, closer inspection of the results and the convergence curves of performance and coverage can help identify a better choice if a clear plateau is observed. To evaluate the stability of the core rule set and automatically identify a more robust set of rules, 100 iterations of a subsampling procedure are used to identify **generalized core rules** (GCRs).”

Reviewer #2:

The work by Klein and colleagues is an intriguing exercise aimed at identifying oncogenic combinations of alterations which characterize a subset of genetically similar tumors. The authors propose a rigorous approach to identify the minimal set of combinations that explain the maximal fraction of patients within a given cohort. In these terms, the authors promote the idea (which I completely espouse) that oncogenic alterations need to be analyzed and considered in a combinatorial manner rather than individually and independently.

The algorithmic approach could be of use to characterize the genetic features of a given tumor cohort, and indeed it would have been nice to have this approach years ago while the TCGA project was running. The results are interesting, although several are known, and the manuscript is well written, even though I would encourage the authors to substantially reduce the use of acronyms and jargon which at times make the text hard to read.

My major concerns are mostly conceptual and on the interpretation of certain results, plus a few methodological clarifications that I feel need to be provided.

1) The main conceptual concern that I have is in the way this method is proposed. The authors begin discussing the relevance of mutually exclusive and cooccurring alterations in cancer, citing the relevant literature (which I'm afraid is actually much vaster, especially for mutual exclusivity) and comparing their approach with a recent algorithm to detect significant mutual exclusivity and cooccurrence.

However, as stated by the authors themselves, their method does not search for statistical significant cooccurrence (see first paragraph pg. 16), but for a minimal set of combinations able to cover the majority of alteration occurrences (in this way concurrent alterations are favored, but the cooccurrence is not necessarily unexpected). This is not a flaw, it could actually be an advantage of this approach, but the comparison with other tools to detect statistically significant cooccurrence is not proper.

This tool is in line with other approaches designed to stratify patients based on minimal subsets of recurrent alterations, see e.g.

<https://www.ncbi.nlm.nih.gov/pmc/articles/PMC4491878/>

<https://www.nature.com/articles/ng.2762>

<https://www.nature.com/articles/nmeth.2651>

<https://pubmed.ncbi.nlm.nih.gov/29949979/>

in all these cases, as in the algorithm presented by the authors, cooccurrence is intrinsically favored by the algorithms but is not assessed per se.

Given the algorithms above have been often applied to the TCGA cohorts (either in their respective publications or directly in the marker studies from TCGA) the authors should

compare their results with results obtained by those approaches and discuss their tool in the context of these and other similar methods.

We thank Reviewer 2 for this comment and for highlighting these 4 papers that are very interesting highly relevant to our study. In order to address this comment, we added the following paragraphs to the discussion section:

“Network-based algorithms have been applied to TCGA datasets to stratify tumors into biological subtypes. Hofree *et al.* developed network-based stratification (NBS), an unsupervised strategy for identifying tumor sub-classes by integrating binary somatic mutation profiles with public gene-gene interaction network databases (Hofree *et al.*, 2013). The authors identified subtypes that are predictive of clinical outcomes in ovarian, uterine and lung cancers. Liu *et al.* applied NBS to a pan-cancer dataset of 12 TCGA cancer types and identified 9 pan-cancer subtypes characterized by different biological processes (Liu and Zhang, 2015). Zhang *et al.* developed a supervised version of NBS that they trained to classify known subtypes of glioblastoma and breast cancer (Zhang *et al.*, 2018). Ciriello *et al.* represented a pan-cancer TCGA cohort as a bipartite graph between samples and selected alterations, and then applied a network-modularity optimization strategy to identify subsets of samples characterized by subsets of alterations (Ciriello *et al.*, 2013).

The aforementioned studies provided functional insights into the biological processes that characterize specific subtypes and identified tumors that are functionally similar at the pathway level. By contrast, CRSO stratifies patients based on specific alteration combinations that cooccur in all of the tumors within a subtype. CRSO and network-based strategies provide complementary information, with network-based approaches elucidating the biological pathways that are perturbed in individual tumors, whereas CRSO can provide insights into the specific alterations that cooperatively enable the pathway dysregulation. Future work can leverage these complementary strategies by applying CRSO within subtypes identified by network-based approaches in order to identify rules that correspond to specific functional subtypes.”

Although all of these methods are similar to CRSO in the sense that they enable discovery of biological subtypes within TCGA datasets, we did not think that a side-by-side comparison of results was within the scope of this paper for several reasons, listed below.

1. The most important obstacle to side-by-side comparison is that the aforementioned network-based approaches do not nominate specific combinations of cooccurring alterations that can be directly compared to the CRSO results. Hofree *et al.* emphasizes that NBS is not concerned with identifying specific cooccurring alterations by stating “we postulated that, although two tumors may not have any mutations in common, they may share the networks affected by these mutations” (Hofree *et al.*, 2013).
2. Qualitative comparison of the results is difficult as well because of key differences in how these studies were structured. Hofree *et al.* only used somatic mutations and did not incorporate CNVs. This is particularly problematic because every core rule identified in ovarian cancer involve at least one CNV, since the somatic alteration landscape in ovarian cancers is dominated by CNVs. Liu *et al.* and Ciriello *et al.*, both applied their methods to pan-cancer datasets, whereas we applied CRSO to individual cancer types (Liu and Zhang, 2015; Ciriello *et al.*, 2013). For reasons that we explain below in comment 3, it would be very difficult to apply the current implementation of CRSO in a pan-cancer cohort. Finally, Zhang *et al.*, performed is difficult to compare with CRSO because they performed a supervised study in which they optimized their algorithm to classify known subtypes based on network modules (Zhang *et al.*, 2018).

We also would like to explain how we addressed Reviewer 2's concern about the appropriateness of comparing CRSO with SELECT, a method based on statistical cooccurrence.

We decided to keep the SELECT comparison even though SELECT and CRSO have different criteria for prioritizing combinations because we think it may be interesting for some readers to have a side-by-side comparison of CRSO results with specific combinations identified with an approach based on statistical cooccurrence. Indeed, SELECT was one of the only methods that we came across that outputted a list of cooccurring alterations hypothesized to cooperate in individual tumors.

However, we made several modifications to the text that we hope will assuage some of Reviewer 2's concerns about this comparison, including the concern described below in comment 2B.

1. We added the following sentence at the end of the SELECT results section:

“Additional factors may contribute to the differences in results between CRSO and SELECT. For example, SELECT combinations were identified across a pan-cancer cohort and then were evaluated within individual cancer types *post-hoc*, whereas CRSO was applied directly to individual cancer types.”

2. We also removed the following sentence from the same paragraph:

“Moreover, the full set of cooccurrences identified by SELECT (i.e., not restricted to common events), included 0 combinations involving *IDH1* in LGG, 0 combinations involving *BRAF* in SKCM, and 1 low-coverage combination involving *NRAS* in SKCM.”

3. To further de-emphasize the comparison with SELECT, we removed the paragraph about the SELECT comparison from the discussion altogether.

We hope that Reviewer 2 will be satisfied with our changes. We tried to preserve a comparison that might be of interest to some readers, while toning down the emphasis on this comparison and explicitly acknowledging that other factors should be considered in evaluating the comparison.

2A) The other major issue I have is with the interpretation of some of the results. There are two strong assumptions in the way the manuscript is currently written which are not substantiated:

First, once a final set of rules is defined for a tumor type, alterations in each patient are classified as driver or passenger based on them being part or not of the rule to which the patient was assigned. I understand this is just nomenclature (or at least I hope it is) but the authors cannot conclude that an alteration in a given patient is non-functional simply because it is not found in the combination predicted by the algorithm. E.g. in figure 4A, several BRAF mutations are classified as passenger: what is the evidence for that?

If the authors can effectively provide data supporting a different functional effect for these mutations, then it would be incredibly interesting to demonstrate that their algorithm, as an added bonus, can indeed discriminate functional and non-functional events.

However, if they can't, the terminology need to be changed and it should made clear in the text that a rule-patient assignment does NOT imply that only alterations in the rule are functional.

We thank Reviewer 2 for bringing this important point to our attention. We eliminated this confusing usage of “passenger” throughout the manuscript, please refer to major change 1 above.

2B) Second, the authors did a commendable effort to survey the literature to find support of functional interactions between concurrent alterations. I'm not sure, however, the combinations detected by this method need to reflect functional interactions. As mentioned in my previous comment, the cooccurrence detected by this tool is not necessarily unexpected (i.e. statistically significant), it might simply reflect tumor subtypes associated with cell of origin (e.g. see the glioma rules, once histological or molecular subtypes are accounted for, the cooccurrence is trivial - which is why they were not found by SELECT), differentiation status or other molecular modifications.

It is in general difficult to establish a chain of causation among the molecular features and phenotypes observed in cancer cells, thus implying that concurrent events in a rule defined by their algorithm are as such because of underlying functional relationships is here over-reaching. The authors should tone down certain statements and acknowledge that several covariates are here not accounted for.

We completely agree that it is over-reaching to infer functional interpretations and relationships based on CRSO results alone, and we have tried to remove unfounded functional assertions throughout the manuscript. We hope that the paragraph we added to the discussion reproduced in “major change 1” helps address this concern as well.

3) It would have been compelling to have this tool running on a cohort combining all tumor types. Was this not done because of efficiency issues? This would have been a great analysis to show how much tumor subtypes can be defined by combination of alterations, oblivious of their sites of origin. Similar analyses have been previously done, it is a pity not having it here.

If indeed efficiency issues prevent this analysis, then I would strongly suggest to include a comparative analysis of rules determined across tumor types, e.g. to propose tumor type specific and pan-tumor rules.

Thank you for this suggestion. To address this comment, we added an analysis of the overlap of the generalized core duos found between different tissues, presented in the new results subsection “Recurrent driver combinations across tissues”, and summarized in a new table, Table 6 (please see “major change 2” above). We think that highlighting duos that were identified in 3 or more cancer types might be very interesting to readers, especially since some unexpected combinations involving lesser-known copy number events.

We chose to analyze the overlap among the individual cancer results instead of performing a single pan-cancer run for a few reasons:

1. Pre-processing the datasets for a pan-cancer run would be difficult and would result in different sets of candidate events. In the presented results, CRSO was applied to each cancer type using published cancer-specific GISTIC2 results from the Broad GDAC

firehose. In order to perform a pan-cancer run, we would have needed to run GISTIC2 ourselves on a pan-cancer cohort. None of us have experience running GISTIC2 directly this pre-processing step could have delayed publication.

2. Beyond the technical difficulties of pre-processing the data for a pan-cancer run, there are conceptual difficulties as well. There is large variation in the number of samples available for different cancer types (ranging from 120 to 963 samples). This variation would lead to prioritization of rules in cancer types with more samples.
3. Moreover, because GISTIC2 searches for genomic regions that are enriched relative to an expectation defined across the full cohort, important candidate CNVs may be left out if they are enriched in under-represented cancer types.
4. The current procedure for calculating passenger-probabilities of CNVs uses the average alteration rates of control cytobands across the cohort as a starting rate for each CNV observation type. The control cytobands themselves are cancer-specific since they are defined to be the full set of cytobands that do not overlap with any of the cancer-specific GISTIC wide peaks. The accuracy of event-specific passenger rates for CNVs is likely to be degraded if calculated using a highly, heterogeneous cohort.

In summary, CRSO is likely to perform better on cohorts that are more homogenous because of the aforementioned limitations in identifying candidate CNVs and calculating event-specific passenger probabilities for CNVs. There are ways to work around these limitations, but they would likely require making major changes to the current pipeline for CNV handling.

We agree with Reviewer 2 that identifying pan-cancer rules is very important and interesting. Because of the above considerations, we decided that the best way to explore pan-cancer rules was to use a bottom-up approach of comparing the results from cancer-specific runs rather than a top-down approach of a single pan-cancer run.

4) From a methodological perspective: what is effect of the scores computed in the P matrix to the final rule set?

By absurd, if all values in P were set to 1 when an alteration occur in a sample (0 otherwise), would the results obtained be dramatically different? I'm asking because the computation of P is quite laborious and might represent a bottleneck to a broad application of this approach, but at the same time the objective function is based on it.

The authors should investigate what is the effect of using simpler versions of P, e.g. having different fixed scores for missense, loss-of-function and hotspot mutations. The way it is currently done is more sophisticated, but how much does it help? It would be important to know.

Thank you for this suggestion. We have added a new section with the suggested experiments: "Comparison of TCGA results using simpler penalty matrices". We describe the new section in "major change 3".

5) On a similar tone, once a core rule set is identified, the approach goes through a lengthy series of refinements. what is the effect of these refinements in terms of result? Do they significantly improve the performance? (coverage? number of rules?)

We Reviewer 2 for this question, and we agree that providing more information about the benefit of the individual phases in CRSO could be of interest to readers. We addressed this question by adding a new figure (Appendix Figure S9) showing the observed benefit of phases 3 and 4 relative to phase 2 in all 19 TCGA cancers. We also added the following text into the Methods subsection “CRSO reports from 19 TCGA cancers”:

“To evaluate the impact of phases 3 and 4, we compared for each tissue the performance of the core RS post phase 2 with the performance of the final core RS, and with the performance of the core RS post phase 3. The mean increases in J and coverage from phase 2 to phase 4 were 14% and 8.3%, respectively (Appendix Figure S9A). The mean increases in J and coverage from phase 2 to phase 3 were 1.2% and 0.6%, respectively (Appendix Figure S9B). Note that the core RS were identical post phase 3 and post phase 2 in 10/19 cancer types.”

The last step of CRSO, after the phase 4 core rule set is identified, is to perform a subsampling that defines a generalized core rule set (GCR). The purpose of this procedure is not to find a rule set that is superior in performance or coverage compared to the core rule set. Rather, the GCR procedure is designed to quantitatively prioritize rules within the core rule set and also to identify rules that were not in the core rule set but were near misses.

Thank you for sending us your revised manuscript. We have now heard back from the two referees who were asked to evaluate your revised study. As you will see below the reviewers think that the study has improved as a result of the performed revisions and are supportive of publication. Reviewer #1 raises a remaining concern, which can be addressed by modifying the Discussion, and we would ask you to address this in a minor revision.

Moreover, we would ask you to address some remaining editorial issues listed below.

REFEREE REPORTS

Reviewer #1:

The authors have dealt with all of my comments except one. The outstanding issue is the possibility of some cancers being driven by a single mutation / SCN. This likely goes some way to explaining (a) why rule sets from CRSO never reach 100% coverage (b) the observation of 'passenger' mutations. This possibility should be added to the new section in the Discussion on the subject of 'passengers' as well as to sections that discuss the coverage of rule-sets.

One other minor typo: 'in in' in the first sentence of the Introduction.

Reviewer #2:

Overall, the authors clarified several of the concerns and introduced new data and analyses that improved the paper.

I'm still not convinced with certain comparisons (which feel like comparing apples and oranges) but the manuscript has certainly gained clarity and in my opinion is ready for publication.

The authors have made all requested editorial changes.

Accepted

26th Jan 2021

Thank you again for sending us your revised manuscript. We are now satisfied with the modifications made and I am pleased to inform you that your paper has been accepted for publication.

Corresponding Author Name: Hongyu Zhao , David F Stern & Michael I. Klein

Manuscript Number: MSB-20-9810R